# SMA-miRs (miR-181a-5p, -324-5p, and -451a) are overexpressed in spinal muscular atrophy skeletal muscle and serum samples

Emanuela Abiusi[1†], Paola Infante[2†], Cinzia Cagnoli[3], Ludovica Lospinoso Severini[4], Marika Pane[5,6], Giorgia Coratti[6], Maria Carmela Pera[6], Adele D'Amico[7], Federica Diano[1], Agnese Novelli[1], Serena Spartano[1], Stefania Fiori[1], Giovanni Baranello[8], Isabella Moroni[9], Marina Mora[9], Maria Barbara Pasanisi[9], Krizia Pocino[10], Loredana Le Pera[11,12], Davide D'Amico[13], Lorena Travaglini[7], Francesco Ria[14,15], Claudio Bruno[16], Denise Locatelli[3], Enrico Silvio Bertini[7], Lucia Ovidia Morandi[9], Eugenio Mercuri[5,6], Lucia Di Marcotullio[4,17‡], Francesco Danilo Tiziano[1,18*‡]

[1]Department of Life Sciences and Public Health, Section of Genomic Medicine, Università cattolica del Sacro Cuore, Roma, Italy; [2]Center For Life Nano Science@ Sapienza, Istituto Italiano di Tecnologia; Department of Molecular Medicine, Università degli Studi di Roma "La Sapienza", Roma, Italy, Roma, Italy; [3]Clinical and Experimental Epileptology Unit, Fondazione IRCCS Istituto Neurologico Carlo Besta, Milano, Italy, Roma, Italy; [4]Department of Molecular Medicine, Università degli Studi di Roma "La Sapienza", Roma, Italy; [5]Pediatric Neurology, Department of Woman and Child Health and Public Health, Fondazione Policlinico Universitario A. Gemelli IRCCS-Università Cattolica del Sacro Cuore, Rome, Italy; [6]Centro Clinico Nemo, Fondazione Policlinico Universitario A. Gemelli IRCCS-Università Cattolica del Sacro Cuore, Roma, Italy; [7]Unit of Neuromuscular and Neurodegenerative Disorders, Dept. Neurosciences, Bambino Gesu' Children's Hospital IRCCS, Roma, Italy; [8]Department of Pediatric Neuroscience, Fondazione IRCCS Istituto Neurologico Carlo Besta, Milano, Italy; [9]Neuromuscular Diseases and Neuroimmunology Unit, Fondazione IRCCS Istituto Neurologico Carlo Besta, Milan, Italy; [10]Department of Medical and Surgical Sciences, Fondazione Policlinico Universitario Agostino Gemelli IRCCS, Roma, Italy; [11]Bioenergetics and Molecular Biotechnologies (IBIOM), CNR-Institute of Biomembranes, Bari, Italy; [12]CNR-Institute of Molecular Biology and Pathology (IBPM), Rome, Italy; [13]Amazentis SA, EPFL Innovation Park, Losanne, Switzerland; [14]Department of Translational Medicine and Surgery, Section of General Pathology, Università Cattolica del Sacro Cuore, Roma, Italy; [15]Fondazione Policlinico Universitario A. Gemelli - IRCCS, Rome, Italy; [16]Center of Translational and Experimental Myology, IRCCS Istituto Giannina Gaslini, Genova, Italy; [17]Laboratory affiliated to Istituto Pasteur Italia-Fondazione Cenci Bolognetti, Department of Molecular Medicine, Sapienza University, Rome, Italy; [18]Unit of Medical Genetics, Department of Laboratory science and Infectious Diseases, Fondazione Policlinico Universitario IRCCS "A. Gemelli", Rome, Italy

*For correspondence: francescodanilo.tiziano@unicatt.it

[†]These authors contributed equally to this work
[‡]These authors also contributed equally to this work

## Abstract

**Background:** Spinal muscular atrophy (SMA) is a neuromuscular disorder characterized by the degeneration of the second motor neuron. The phenotype ranges from very severe to very mild forms. All patients have the homozygous loss of the *SMN1* gene and a variable number of *SMN2* (generally 2–4 copies), inversely related to the severity. The amazing results of the available treatments have made compelling the need of prognostic biomarkers to predict the progression trajectories of patients. Besides the *SMN2* products, few other biomarkers have been evaluated so far, including some miRs.

**Methods:** We performed whole miRNome analysis of muscle samples of patients and controls (14 biopsies and 9 cultures). The levels of muscle differentially expressed miRs were evaluated in serum samples (51 patients and 37 controls) and integrated with *SMN2* copies, *SMN2* full-length transcript levels in blood and age (SMA-score).

**Results:** Over 100 miRs were differentially expressed in SMA muscle; 3 of them (hsa-miR-181a-5p, -324-5p , -451a ; SMA-miRs) were significantly upregulated in the serum of patients. The severity predicted by the SMA-score was related to that of the clinical classification at a correlation coefficient of 0.87 ($p < 10^{-5}$).

**Conclusions:** miRNome analyses suggest the primary involvement of skeletal muscle in SMA pathogenesis. The SMA-miRs are likely actively released in the blood flow; their function and target cells require to be elucidated. The accuracy of the SMA-score needs to be verified in replicative studies: if confirmed, its use could be crucial for the routine prognostic assessment, also in presymptomatic patients.

**Funding:** Telethon Italia (grant #GGP12116).

## Introduction

Spinal muscular atrophy (SMA) is an autosomal-recessive neuromuscular disorder, characterized by the degeneration of the α-motor neurons of the ventral horns of the spinal cord. The severity of the infantile forms is classically ranked into types I–III, according to the age of onset and the maximum motor milestone achieved (*Mercuri et al., 2020*). According to the conventional classification, the onset of the condition in type I is below 6 months of age and patients do not acquire the sitting position; in type II, symptoms occur within 18 months and children do not acquire autonomous ambulation. Type III is the most variable phenotype; developmental phases are comparable to that of the general population, the onset is over 18 months. Type III is subclassified into -a and -b, based on onset below or over 3 years of age (*Zerres and Rudnik-Schöneborn, 1995*). An additional form, with onset over 18 years of age, is usually reported as type IV: the outcome is variable and the degree of disability is usually mild. Indeed, SMA phenotype is better depicted as a continuous spectrum: for this reason, patients are preferably stratified according to a decimal classification, available for types I and II only (*Dubowitz, 1995*; *Main et al., 2003*).

Irrespective of the phenotypic severity, SMA patients have the same genetic defect: the homozygous loss of the *SMN1* gene, located in 5q13 (*Lefebvre et al., 1995*). In the same region, a hypomorphic allele of *SMN1* is present (*SMN2*), which produces insufficient levels of the SMN protein. Due to an alternative splicing, *SMN2* is mainly transcribed into an isoform lacking exon 7 (SMN-del7) and then translated into an unstable protein (*Mercuri et al., 2020*). The number of *SMN2* genes is variable in patients, generally 2–4; *SMN2* copy number is the only consistent phenotypic modifier known to date, grossly and inversely related to disease severity (*Calucho et al., 2018*). The efficiency of exon 7 inclusion in *SMN2* mRNA can be enhanced by two relatively rare variants (rs121909192 and rs1454173648), more commonly found in the less severely affected patients (*Vezain et al., 2010*; *Wu et al., 2017*).

The molecular pathophysiology of SMA is largely unknown: even though the SMN protein is ubiquitously expressed and has a housekeeping function in splicing regulation, the second motor neuron is the main target cell of the disease (*Beattie and Kolb, 2018*). However, a growing bulk of evidence supports the pathogenic role of skeletal muscle or even of the whole motor arch (*Bricceno et al., 2014*; *Martínez-Hernández et al., 2014*; *Ripolone et al., 2015*).

Physiological and pathological degeneration of skeletal muscle in SMA has been intensively investigated, also focusing on the role of microRNAs (miRs or miRNAs), 20-22mers non-coding RNAs. miRs are involved in most cellular processes: these regulate gene expression through mRNA degradation or translational inhibition, mainly by binding cis-regulatory elements present in the 3′UTR of mRNAs (*Bartel, 2009*; *Vidigal and Ventura, 2015*). Among miRs, the myomiRs play a pivotal role in regulating myogenesis and muscle degeneration (including hsa-miR-1, miR-206, miR-133a, and miR-133b) (*Horak et al., 2016*).

miRs have been intensively studied also as potential biomarkers in several conditions, including SMA (*Kariyawasam et al., 2019*). This topic has become even more relevant after the development and registration of the first effective treatments, which have led to a revolutionary change of perspective for patients (*Mercuri et al., 2020*). The results of presymptomatic treatments (*Mercuri et al., 2020*) are prompting the development of newborn screening programs, already started in some countries worldwide (*Dangouloff et al., 2020*), and are making compelling the need of prognostic biomarkers to predict the progression trajectories of patients identified at birth. Besides the *SMN2* products, either transcripts or protein, (*Tiziano et al., 2013*; *Tiziano et al., 2019*; *Crawford et al., 2012*), few SMN-independent biomarkers have been evaluated so far, with discordant results; these include the SMA-MAP, creatinine, neurofilament dosage, and a few miRs (*Kariyawasam et al., 2019*; *Kobayashi et al., 2013*; *Alves et al., 2020*; *Darras et al., 2019*).

In the present study, we have first used whole miRNome sequencing of muscle specimens and cultures from SMA patients and controls to identify a specific signature of the disease and investigate on the muscle involvement in the pathogenic process. Subsequently, the levels of deregulated miRs have been evaluated in serum samples of patients and controls as SMN-independent biomarkers.

We identified three deregulated miRs (miR-181a-5p, miR-324-5p , miR-451a: SMA-miRs) that have been integrated in a composite score (SMA-score), including *SMN2* full-length (*SMN2*-fl) transcript levels, *SMN2* copy number and age at sampling, markedly improving the phenotypic predictive value of *SMN2* copy number assessment alone.

# Methods

**Key resources table**

| Reagent type (species) or resource | Designation | Source or reference | Identifiers | Additional information |
|---|---|---|---|---|
| Gene (*Homo sapiens*) | *SMN1* | GenBank | HGNC:HGNC:11,117 | |
| Gene (*Homo sapiens*) | *SMN2* | GenBank | HGNC:HGNC:11,118 | |
| miR (*Homo sapiens*) | hsa-miR-181a-5p | miRBase | MIMAT0000256 | |
| miR (*Homo sapiens*) | hsa-miR-324-5p | miRBase | MIMAT0000761 | |
| miR (*Homo sapiens*) | hsa-miR-451a | miRBase | MIMAT0001631 | |
| Strain, strain background (*Mus musculus*) | SMNΔ7 mice FVB.Cg-Grm7Tg(SMN2)89Ahmb Smn1tm1Msd Tg(SMN2*delta7)4,299Ahmb/J | Jackson Laboratory | Stock number: 005025 | Hum Mol Genet 14(6):845–57, 2005 |
| Genetic reagent (*Homo sapiens*) | miRCURY LNA microRNA mimic: hsa-miR-181a-5p, hsa-miR-324-5p , hsa-miR-451a | Exiqon | | 50–100–200 nM |
| Genetic reagent (*Homo sapiens*) | miRCURY LNA microRNA antogomiR: hsa-miR-181a-5p, hsa-miR-324-5p , hsa-miR-451a | Exiqon | | 0.1 nM |
| Cell line (*Homo sapiens*) | Primary myoblasts | Italian Telethon Network of Genetic Biobanks | 6756, 6760, 6762, 6816, 7147, 8823, 8655, 8537 | |
| Biological sample (*Homo sapiens*) | Muscular biopsies | Italian Telethon Network of Genetic Biobanks | 10370, 10351, 8023, 4688, 10543, 10583, 7669, 5944, 5824, 5717, 6760, 6438, 6082, 5379, 7689, 5842, 9814 | |
| Sequence-based reagent | See *Supplementary file 3* | IDT (Integrated DNA Technologies) | | |

*Continued on next page*

*Continued*

| Reagent type (species) or resource | Designation | Source or reference | Identifiers | Additional information |
|---|---|---|---|---|
| Commercial assay or kit | TruSeq Small RNA Sample Preparation kit | Illumina | TruSeq Small RNA Library Prep Kit – RS-200-0024 | |
| Commercial assay or kit | miRCURY RNA Isolation Kit – Biofluids | Exiqon | | |
| Commercial assay or kit | Universal cDNA synthesis kit II | Exiqon | | |
| Commercial assay or kit | Pick-&-Mix miRNA PCR Panel 96-well | Exiqon | | |
| Commercial assay or kit | E.Z.N.A PX Blood RNA Kit | Omega bio-tek | SKU: R1057-01 | |
| Commercial assay or kit | High Capacity cDNA Reverse Transcription Kit | Thermo Fisher Scientific | Catalog number: 4368814 | |
| Software, algorithm | Illumina Genome Analyzer | Illumina | | |
| Software, algorithm | RealTime StatMinerVersion 4.1 | | | |
| Software, algorithm | Statgraphics Centurion XV software | StatPoint Inc | | |
| Software, algorithm | SPSS 18.0 software | SPSS | RRID:SCR_002865 | |

## Samples

Muscle biopsies from seven SMA patients (three SMA I, two SMA II, and two SMA III) were obtained from the Italian Telethon Network of Genetic Biobanks, held at Istituto Neurologico Carlo Besta in Milan. The characteristics of subjects are specified in *Supplementary file 1*. The selection criteria of specimens were as follows: (1) unique site of sampling (femoral quadriceps) and (2) first stage of disease (defined as onset of the first clinical signs that prompted the diagnostic workflow) to minimize the presence of fibrosis. Seven muscle biopsies were selected as controls among morphologically normal samples of age-matched subjects who underwent muscular biopsy for neonatal hypotonia (for type I) or hyper-CKemia (for types II and III).

Muscle cell cultures (five from SMA patients and four from controls) were obtained from the Telethon Biobank or set in-house, as previously reported (*Zanotti et al., 2007*). Four patients (two type I and two type II) had the homozygous deletion of *SMN1*; the latter was a type I subject, compound heterozygote for the c.439_443delGAAGT (p.Glu147SerfsTer2) variant.

51 SMA patients were enrolled (3 type I, 21 type II, 26 type III, 1 type IV; *Supplementary file 2*). 47 DNA samples were available for *SMN2* copy number assessment and SNP (rs121909192 and rs1454173648) genotyping; whole blood samples were collected from 45 patients for total RNA extraction, and 51 serum samples were available for small RNA extraction.

Serum samples from 37 controls from the general population were analyzed: the selection of these subjects was performed according to a pairing criterion for age and sex with patients. Samples were obtained upon anonymization from the discards of the routine analysis laboratories of Fondazione Policlinico Gemelli and Ospedale Pediatrico Bambino Gesù; only information on sex and age was retained. From controls, no DNA or total RNA were extracted. The two groups were homogenous for sex (28 and 19 females, respectively) and age (mean $17.3 \pm 19.2$ and $11.9 \pm 12.4$ years, respectively, $p > 0.05$).

This study was approved by the Ethics Committee of Fondazione Policlinico Universitario IRCCS "A. Gemelli" (Prot # 25188/19, ID: 2614).

For SMA mice, the original breeding pairs of SMNΔ7 mice were purchased from Jackson Laboratory (stock number 005025). The colony was maintained by interbreeding carrier mice, and the offspring were genotyped by PCR assays on tail DNA according to the protocols provided by Jackson Laboratory. According to the ARRIVE guidelines, procedures were carried out to minimize discomfort and pain in compliance with national (D.L. 116 Suppl 40/1992 and D.L. 26/2014) and international guidelines and laws (2010/63/EU Legislation for the protection of animals used for scientific purposes). The experimental protocols were approved by the Ethics Committee of the Fondazione IRCCS Istituto Neurologico C. Besta and by the Italian Ministry of Health (protocol numbers: 962/2016-PR and 1039/2020-PR).

## Cell cultures

Myoblasts (passages from 5 to 15) were cultured in high-glucose Dulbecco Modified Eagle's Medium (DMEM), 20% fetal bovine serum (FBS), 100 u/ml of penicillin, 100 mg/ml of streptomycin, 2 mM L-glutamine, 10 ng/ml of epidermal growth factor (EGF), and 10 μg/ml of bovine insulin in 5% $CO_2$ atmosphere. For myotube differentiation, we have used a standard protocol of serum deprivation (5% FBS) for 2 weeks.

SH-SY5Y (human neuroblastoma) were already present in-house for previous studies and were tested for mycoplasma contamination. We verified the identity of SH-SY5Y cell line by evaluation of neuronal phenotype, following differentiation with 10 pM retinoic acid for 7 days. Cells were cultured in 1:1 DMEM/Ham's F12 nutrient medium with 20% FBS, 100 u/ml of penicillin, 100 mg/ml of streptomycin, and 2 mM L-glutamine.

## Patients

The patients included in the present study were in routine clinical follow-up in the four participating Italian referral neuromuscular centers (Fondazione Policlinico Universitario IRCCS "A. Gemelli," Istituto Neurologico IRCCS "Carlo Besta," Ospedale Pediatrico Bambino Gesù, Istituto Giannina Gaslini). Subjects were evaluated by expert neurologists/pediatric neurologists/physiotherapists: SMA type was first attributed to each patient, according to the usual classification (types I–IV). Three of us (MP, GC, EM) were requested to assign each patient to a SMA subtype, based on the clinical data, according to the decimal classification for types I and II (*Zerres and Rudnik-Schöneborn, 1995*; *Dubowitz, 1995*). For type III, due to the lack of a decimal classification, we arbitrarily assigned the value 3 to type IIIa and 3.5 to type IIIb.

## Whole miRNome sequencing

Total RNA from muscle biopsies and myoblast/myotube cultures was extracted by TRIzol Reagent (Life Technologies) as specified in the manufacturer's protocol. Libraries were obtained through the *TruSeq Small RNA Sample Preparation kit* (Illumina). Next-generation sequencing (NGS) miRNome analysis was performed by Illumina Genome Analyzer (GAIIX) platform.

For miRNome analysis, we have used the following pipeline. The sequencing raw data (.bcl files) were processed by the Illumina Casava software (v1.8.0) to convert the data into fastq files (raw data of miRSeq have been deposited at NCBI-SRA database; BioProject PRJNA748014). The fastq files (31-base single-end reads) were first filtered by quality using the FASTX-toolkit (fastq_quality_filter: -q28 p50) and then trimmed to remove the adapter from their 3′ end (TrimGalore tool). Only reads longer than 15 bases were retained and mapped on the miR-precursor sequences annotated in the miRBase repository (v19). The Bowtie2 algorithm was used for the alignment, allowing no more than two mismatches. Quantification, TMM normalization, and significant differential expression test of the known mature miRs were performed using the edgeR package (v2.4.1). Only miRs with >1 count per million (cpm) in at least one condition and in a minimum number of samples (depending on group size) were retained. Multiplicity correction was performed by applying the Benjamini–Hochberg method on the p-values to control the false discovery rate (FDR). The significantly up- and downregulated miRs were selected at FDR < 0.05.

## Molecular biomarkers

Genomic DNA was extracted from whole blood by conventional salting out method. *SMN2* copy number and RNA analyses were carried out as previously reported (*Tiziano et al., 2010*). The presence of the exon 7 splicing modifier variants (rs121909192 and rs1454173648; *Vezain et al., 2010*; *Wu et al., 2017*) was assessed by Sanger sequencing with the R111 primer (*Lefebvre et al., 1995*), following PCR amplification with primers R111 and C1120 (*Lefebvre et al., 1995*).

Whole blood was collected in PAXgene blood RNA tubes (BD Biosciences) and total RNA extracted by the *E.Z.N.A. PX Blood RNA Kit* (Omega bio-tek), according to the manufacturer's protocol. RT-PCR was performed by the *High Capacity cDNA Reverse Transcription Kit* (Thermo Fisher Scientific). *SMN2* transcript levels were assessed as previously reported (*Tiziano et al., 2010*).

miRs were extracted from serum by *miRCURY RNA Isolation Kit – Biofluids* (Exiqon). RT-PCR was performed by *Universal cDNA synthesis kit II* (Exiqon). For both protocols, manufacturer's instructions were followed.

Commercial relative qPCR assays were purchased at Exiqon (*Pick-&-Mix miRNA PCR Panel 96-well*); UniSp6 was used as calibrator (*miRCURY LNA primers*). miRs with Ct <34 were considered as expressed.

For absolute qPCR assays, specific forward primers were designed based on the target mature miR sequence reported in http://www.mirbase.org. The reverse primer was shared by all assays and was complementary to the tag of the *Universal cDNA synthesis kit II*. The melting temperature (Tm) of primers was established by the *OligoAnalyzer 3.1 tool* (available at the Integrated DNA Technologies website, http://www.idtdna.com). In case of forward oligo Tm <60 °C, the latter was optimized by adding a $(GACT)_n$ tail at the 5' end. For external standard construction, we proceeded as previously described (*Tiziano et al., 2013*). Amplicons for cloning were obtained by PCR-filling of two partially overlapping sequences (*Supplementary file 3*).

## *In silico*, in vitro, and in vivo experiments

miR-181a-5p, miR-324-5p , miR-451a mimics and scramble, and the respective antagomiRs were purchased at Exiqon (*miRCURY LNA microRNA mimicS/antogomiR*). For in vivo experiments, antagomiRs were resuspended in artificial cerebrospinal fluid (aCSF: NaCl 119 mM, $NaHCO_3$ 26.2 mM, KCl 2.5 mM, $NaH_2PO_4$, 1 mM, $MgCl_2$ 1.3 mM, glucose 10 mM).

SH-SY5Y cells were transfected with three different concentrations (50–100–200 nM) of miR-181a-5p, miR-324-5p , miR-451a mimics or scramble. Transient transfections were performed through Lipofectamine 2000 (Invitrogen), according to the manufacturer's protocols. Both total and small RNAs were extracted by the *miRvana Paris RNA extraction kit* (Ambion). RT-PCR was performed by *High Capacity cDNA Reverse Transcription Kit* (Applied Biosystems) or *Universal cDNA synthesis kit II* (Exiqon) for mRNAs or miRs, respectively. miR-181a-5p, miR324-5p, miR-451a, and *SMN2* transcript (*SMN-fl*, *SMN-del7*) levels were quantified as described above.

At postnatal day 1 (P1), SMA-like pups (Smn−/−, hSMN2+/+, SMNΔ7+/+) were cryo-anesthetized and injected with 5 µl of 0.1 nmol of each specific antagomiR, into the cerebral lateral ventricle. Injections were performed with a pulled capillary needle under the guidance of a transilluminator as reported (*Glascock et al., 2011*) All the litters were culled so that each litter contained six siblings, daily weighted, and controlled. miRwalk 3.0 (*Dweep et al., 2014*) was used to identify miRs binding the 3'-UTR of the *SMN2* genes.

## Sequencing analysis of SMA-miR genes

We have amplified the genomic regions of interest by PCR in a final volume of 12.5 µl using the 2X GoTaq Hot Start Colorless Mastermix (Promega) and 0.4 µM of each primer pair (*Supplementary file 3*). The amplification cycle was : 95 °C 5'; (95 °C 45''; 60 °C 45''; 72 °C 30'') × 35; 72 °C 5'; 4 °C. Thereafter, following purification of PCR products by ExoSap-IT (USB Corporation), sequencing reactions were performed by the BigDye Terminator v3.1 Cycle Sequencing Kit (Applied Biosystems) and purified by the BigDye XTerminator Purification Kit (Applied Biosystems). DNA sequencing was performed by capillary electrophoresis using the ABI-Prism 3130 instrument (Applied Biosystems). Electropherograms were analyzed with the *Sequencing Analysis Software 6* (Applied Biosystems).

## Statistical analysis

Relative qPCR data were analyzed using RealTime StatMinerVersion 4.1 software and Benjamini–Hochberg FDR method. Grubbs' test was employed to exclude the outliers. miR levels in patients and controls were compared by non-parametric tests (Wilcoxon test); miRs were identified as significantly differentially expressed at FDR <0.05.

Absolute qPCR data were analyzed by Statgraphics Centurion XV software (StatPoint Inc). miR levels in patients and controls were compared by Mann–Whitney U test, setting α value at 0.05. While in relative qPCR a large number of miRs were analyzed simultaneously, in absolute qPCR experiments the levels of each miR were evaluated separately; for this reason, the FDR threshold was not applied. To rule out possible false-positive results, we opted for increasing the number of samples in each group only for significant miRs (p<0.05) or showing a trend of significance.

For clinical and molecular correlations, continuous variables were compared by linear regression models. Multiple regression analysis was used to correlate clinical severity and molecular parameters (miR levels, *SMN2* transcripts, *SMN2* copy number), setting SMA type as dependent variable. Receiver

operating characteristic (ROC) curves were constructed by SPSS 18.0 software; the cutoff value of single miR or their sum was identified according to the highest values of sensitivity and specificity.

The survival analysis in SMA-like mice was made with SPSS 18.0 software; treated and untreated mice were compared by Kaplan–Meier survival curves; differences in survival were estimated by the log-rank test.

For all tests, p≤0.05 was considered significant.

## Results

### miRNome profile suggests a primitive muscular defect in SMA patients

The analysis of the whole miRNome of muscle biopsies showed a distinct clusterization of patient and control samples (*Figure 1A*). Similar findings were obtained also for myoblast and myotube cultures (*Figure 1B–C*). Globally, miR production was preserved; at $\alpha \leq 0.05$, 99, 20, and 19 miRs were differentially expressed in SMA biopsies, myoblast, and myotube cultures, respectively (*Supplementary file 4*).

The three groups of samples shared five upregulated miRs (hsa-miR-1, -133a , -133b, -204-5p , -208b, *Figure 1D*), mostly belonging to myomiRs (except for hsa-miR-204-5p ). Three differentially expressed miRs were in common between myoblasts and myotubes (hsa-miR-206, -483-5p , and -4697-3p ); two were shared between myoblasts and biopsies (hsa-miR-146a-5p and -184), and three between myotubes and biopsies (hsa-miR-378a-3p, -378f , -501-5p ) (*Figure 1D*); hsa-miR-378a-3p and -378f  had opposite trend (upregulated in myotubes, downregulated in biopsies).

### Serum levels of miR-181a-5p, miR-324-5p, and miR-451a are candidate biomarkers for SMA

Based on the data above, we determined serum levels of miRs that were differentially expressed in muscle samples of patients to identify potential SMN-independent biomarkers for SMA. The validation pipeline is schematized in *Figure 2*; the results are summarized in *Supplementary file 5*. Briefly, as a first-tier test we determined the levels of the 99 deregulated miRs in serum samples from 10 patients (one SMA I, 9 SMA II; median age 1.8 years;  five females) and 10 age-matched controls. This subgroup of patients has been selected by homogeneity of age and severity. We included 11 additional miRs that were identified in SMA patients in other studies (*Kye et al., 2014*; *Valsecchi et al., 2015*; *Murdocca et al., 2016*; *Catapano et al., 2016*; *Wertz et al., 2016*; *O'Hern et al., 2017*; *Sison et al., 2017*; *Kaifer et al., 2019*; *Bonanno et al., 2020*; *Haramati et al., 2010*; *Gonçalves et al., 2018*; *Kirby and McCarthy, 2013*) or with key function in skeletal muscle. For 74 miRs, qPCR assays were commercially available: the 24 miRs that were differentially expressed were validated in a larger cohort. Globally, we developed in-house absolute qPCR assays for 60 miRs: 24 for the miRs to be validated from the first-tier test, and 36 for the remaining.

The validation step has been performed in 51 patients (3 SMA I, 21 SMA II, 26 SMA III, 1 SMA IV) and 37 age- and sex-matched controls (*Supplementary file 2*). Most miRs were undetectable in serum samples or did not display different expression levels in the two groups. Three miRs were significantly upregulated in SMA patients: miR-181a-5p, miR-324-5p,  and miR-451a (SMA-miRs; Mann–Whitney U test, $p = 4.3 * 10^{-4}$; 0.02; 0.004, respectively; *Figure 3A–C*). To rule out that the observed differential expression could be biased by RNA quality/quantity, we performed linear correlation analysis among the three miRs: miR-181a-5p and miR-451a levels were related in both patients and controls, while miR-324-5p  levels were independent from the two others (*Figure 3—figure supplement 1*).

We evaluated sensitivity and specificity of the quantification of the SMA-miRs by using the ROC curves: the highest predictive value was found for miR-181a-5p, with 75% and 61% of sensitivity and specificity, respectively (cutoff 70.5 molecules/µl of serum, $p < 10^{-4}$, *Figure 3D*). To evaluate whether the combination of the SMA-miRs could be more predictive than the levels of the single miR, we constructed the ROC curves of the sum and found an increase in both sensitivity and specificity, up to 80 and 75%, respectively (cutoff 380 molecules/µl of serum, $p < 10^{-4}$, *Figure 3D*).

Globally, the SMA-miRs did not show any correlation with age or sex (p>0.05, *Figure 3—figure supplements 1 and 2*); in patient samples, miR-181a-5p levels were significantly increased in females compared to males (p=0.024, *Figure 3—figure supplement 2*).

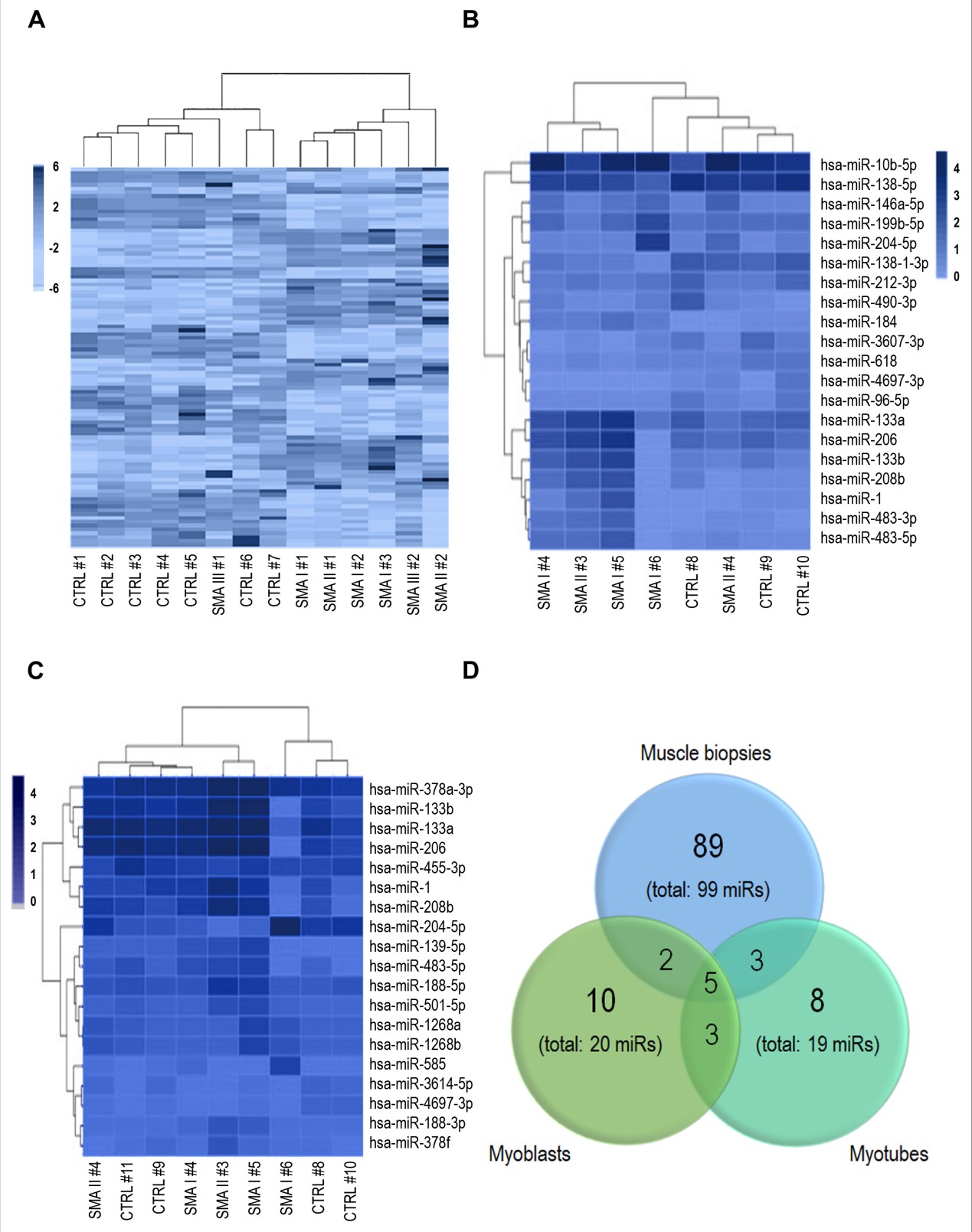

**Figure 1.** Heatmaps obtained by the whole miRNome analysis of muscle biopsies (**A**), myoblasts (**B**), and myotubes (**C**) of spinal muscular atrophy (SMA) patients and controls; patient and control samples display a separate clusterization. 99, 20, and 19 miRs were found deregulated in SMA in muscle biopsies, myoblasts, and myotubes, respectively; (**D**) Venn's diagram showing the five miRNAs shared among the three groups, three between myoblasts and myotubes, two between myoblasts and biopsies, and three between myotubes and biopsies.

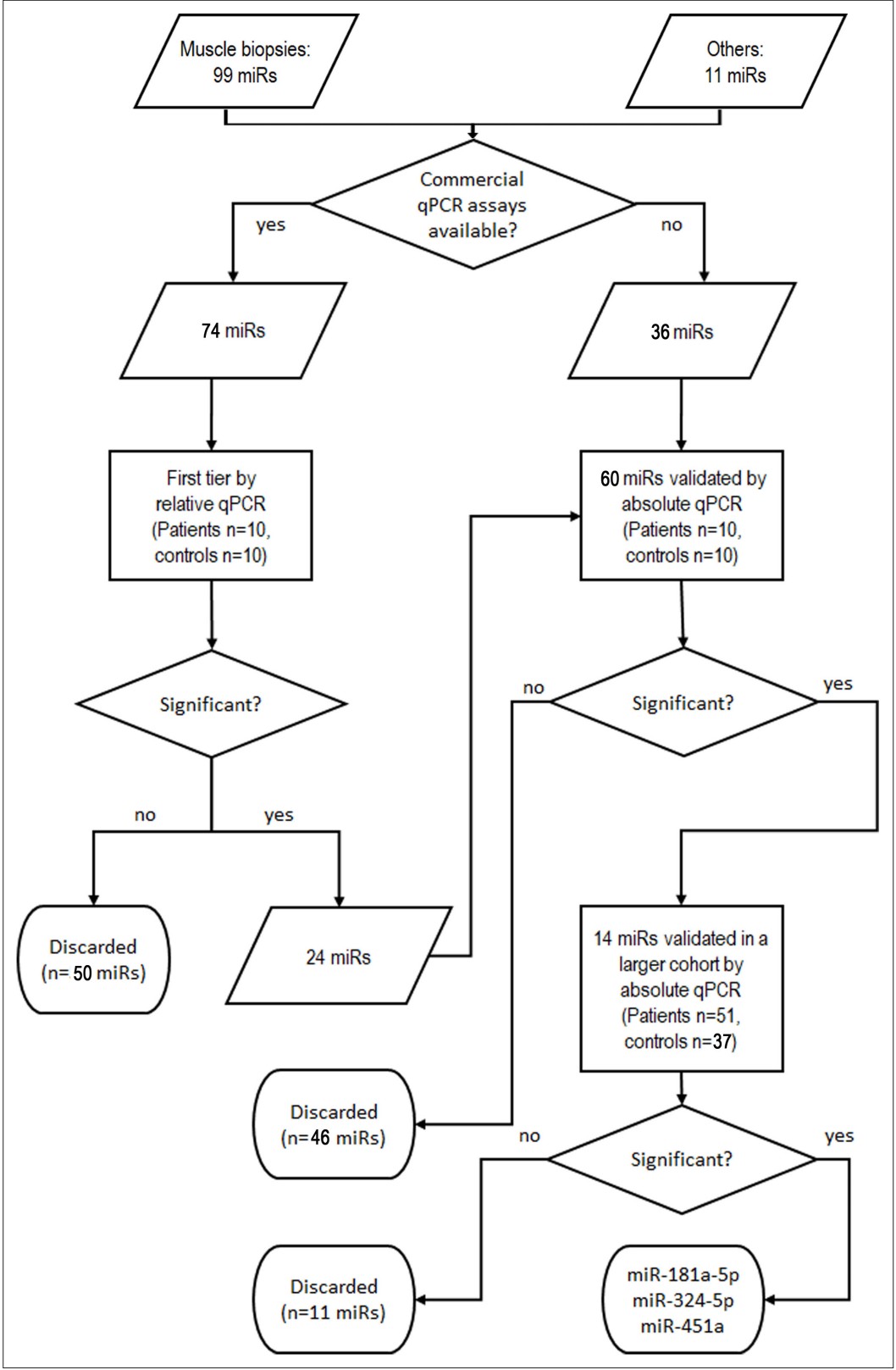

**Figure 2.** Validation pipeline of miRNAs identified by whole miRNome analysis in serum samples of patients and controls. 'Others' indicates miRs that were identified in other studies or with key function in skeletal muscle.

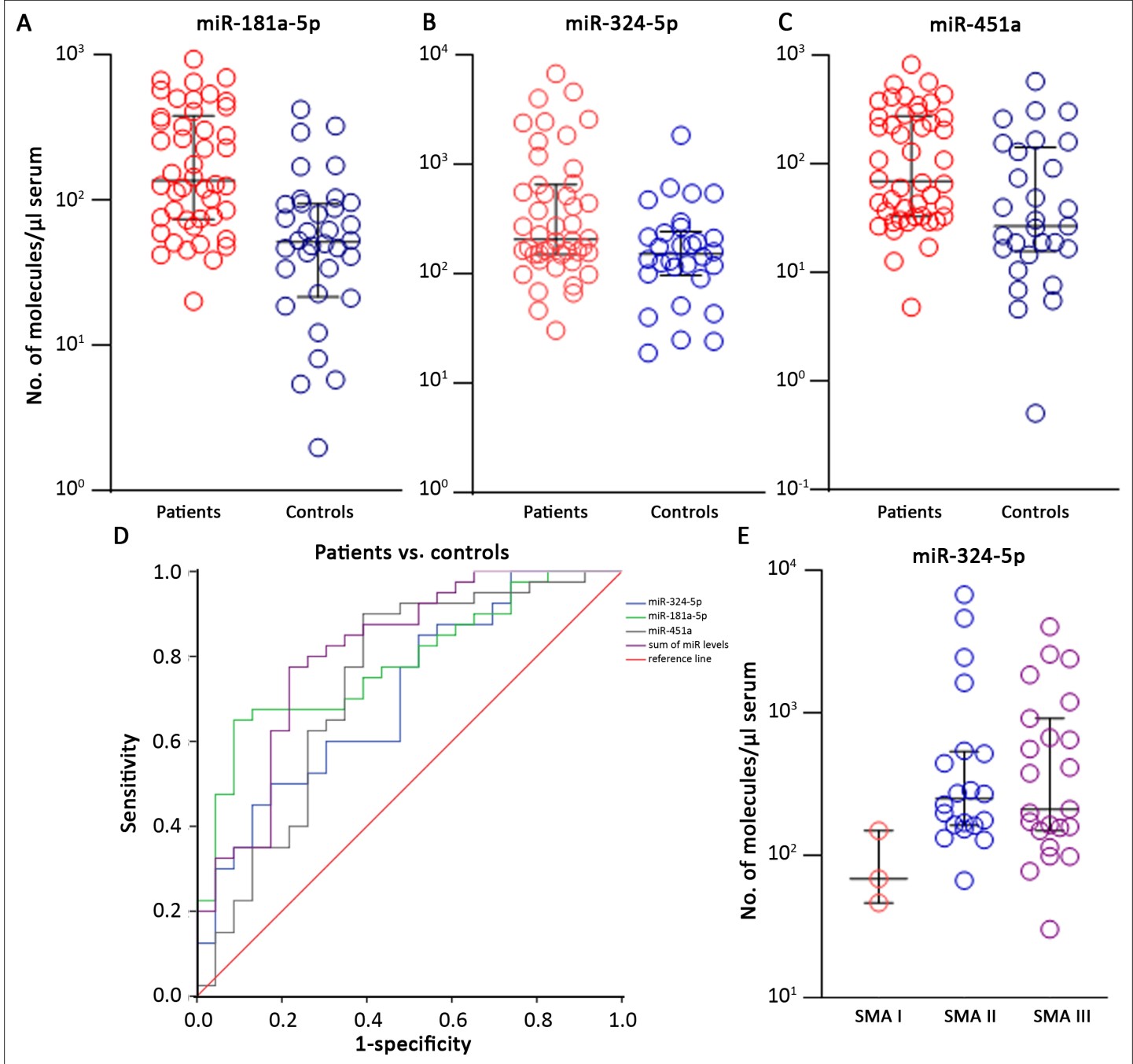

**Figure 3.** The SMA-miRs (miR-181a-5p [**A**], miR-324-5p [**B**] and miR-451a [**C**]) were significantly upregulated in serum samples of spinal muscular atrophy (SMA) patients (p=4.3 * 10⁻⁴; 0.02; 0.004, respectively). Receiver operating characteristic (ROC) curves showed that the quantification of SMA-miRs has 80% sensitivity and 75% specificity in distinguishing patients from controls (**D**). Correlation of miR-324-5p with SMA type (**E**): the levels in SMA II and SMA III patients were significantly increased compared to those of SMA I patients (p=0.03 and 0.04, respectively).

The online version of this article includes the following figure supplement(s) for figure 3:

**Figure supplement 1.** Multiple variable correlation of miR-181a-5p, -324-5p, and -415a levels and age at sampling.

**Figure supplement 2.** Comparison of levels of miR-181a-5p, -324-5p, and -451a in male and female patients; only miR-181a-5p showed a significant difference in females compared to males (*p=0.024).

**Figure supplement 3.** Analysis of type II and III patients with three *SMN2* copies; the two groups were not different for miRs levels (p>0.05, **A**) but showed a significant difference in age (**p=0.0092, **B**).

**Figure supplement 4.** Transfections of SH-SY5Y neuroblastoma cells with SMA-miR mimics (final concentration: 50 or 100 nM).

Regarding the correlation between SMA-miR levels and the severity of the disease, only the levels of miR-324-5p were significantly decreased in type I compared to types II and III (p=0.03 and 0.04, respectively; *Figure 3E*). Finally, we compared miR levels in type II and III patients with three *SMN2* copies. The difference in miR levels was not significant; however, the two groups were significantly different by age (p=0.0092; *Figure 3—figure supplement 3A and B).*

### SMA-miRs do not modulate SMN transcript levels

To evaluate whether SMA-miRs could modify *SMN1* or *SMN2* expression levels, we performed transient transfections of SH-SY5Y neuroblastoma cells with commercial mimics or scramble. The transfection of the single mimics, at the final concentration of 50 or 100 nM, led to an increase of SMA-miR levels from 0 to 1 mol/ng of total RNA in untreated cells, up to 60–100 or 160–200 mol/ng, respectively (*Figure 3—figure supplement 4A*). Despite the huge increase in SMA-miR levels, *SMN1/SMN2* transcripts remained unchanged, except for the SMNdel7 isoform in cells treated with miR-324-5p , which was reduced by 50% independently of the mimic concentration (*Figure 3—figure supplement*

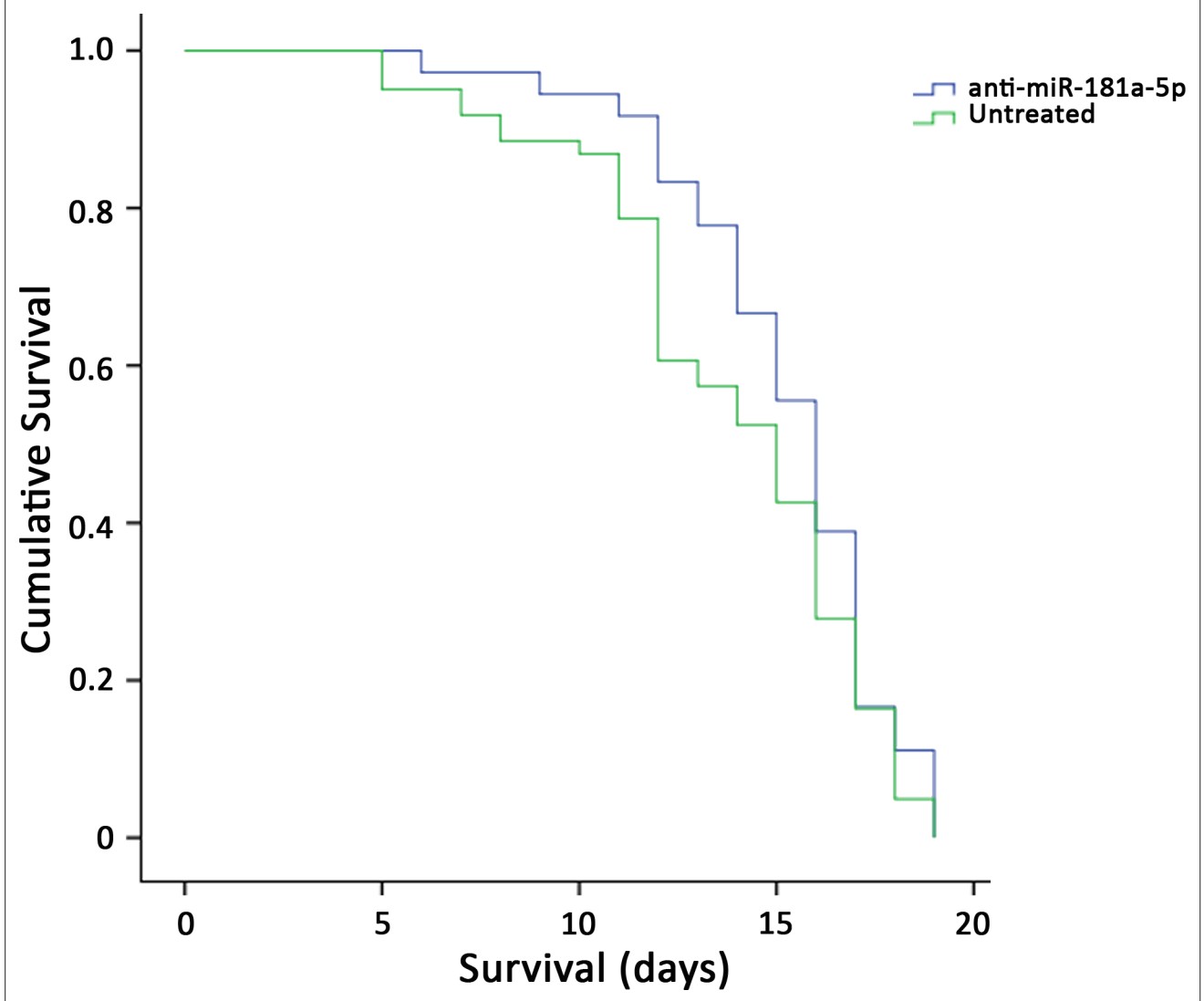

**Figure 4.** Survival curves of SMNΔ7-mice treated with intrathecal injection of anti-miR-181a-5p (n = 36) and untreated (n = 71); the overall survival remained unchanged (p>0.05).

The online version of this article includes the following figure supplement(s) for figure 4:

**Figure supplement 1.** SMNΔ7 mice treated with anti-miR-324-5p  showed a significant transient increase in body weight, between P7 and P10 (p=0.002).

**Table 1.** miRs differentially expressed in spinal muscular atrophy (SMA), as reported in previous studies.

| hsa-miR | miRNome | Relative qPCR | Absolute qPCR* | Reference |
|---|---|---|---|---|
| miR-19a-3p | Upregulated | Upregulated | <10 molecules/µl serum; p=0.42 | *Haramati et al., 2010*; *Gonçalves et al., 2018* |
| miR-23a-3p | Downregulated | Upregulated | 150–200 molecules/µl serum; 10 patients/controls analyzed; p=0.62 | *Kaifer et al., 2019* |
| miR-206 | Nonsignificant | Upregulated | 50–100 molecules/µl serum; 15 patients/controls analyzed; p=0.24 | *Valsecchi et al., 2015*; *Catapano et al., 2016*; *Bonanno et al., 2020* |
| miR-9 | Nonsignificant | Not tested | <10 molecules/µl serum; p=0.30 | *Catapano et al., 2016* |
| miR-132 | Nonsignificant | Not tested | Not tested | *Catapano et al., 2016* |
| miR-146a | Upregulated | Upregulated | <5 molecules/µl serum; p=0.10 | *Sison et al., 2017* |
| 4miR-431 | Nonsignificant | Not tested | Not tested | *Wertz et al., 2016* |
| miR-183 | Nonsignificant | Not tested | Not tested | *Kye et al., 2014* |

*p-Values refer to the significance of comparison of the miR levels in patients and controls by Mann–Whitney U-test. p-values < 0.05 were considered significant.

*4B*). Since in untreated cells SMA-miR levels were almost undetectable, we did not perform experiments with the antagomiRs.

## SMA-miRs, in the absence of SMN-modifying treatments, do not improve the survival of SMA-like mice

To test the hypothesis of the retrograde effect of the secretion of SMA-miRs on spinal cord cells, we evaluated the survival of SMNΔ7 mice in the absence of any modification of SMN levels. We first treated at P1 five affected mice by intrathecal injection of each specific SMA-miR antagonist. Anti-miR-324-5p and anti-miR-451a did not affect mice survival and were not further studied, although the first led to a significant transient increase in body weight (between P7 and P10, p=0.002; *Figure 4—figure supplement 1*). Conversely, since anti-miR-181a-5p significantly improved the survival, we increased the cohort of treated animals (n = 36), which were compared with untreated animals (n = 71), and with those treated with the scramble (n = 31). The overall survival remained unchanged (p>0.05, *Figure 4*).

## SMA-miRs do not display gene variants in patients

We tested whether sequence variation in genes encoding the SMA-miRs could be related to SMA phenotype. We did not identify any variant in 38 samples of patients affected from different forms of SMA (3 type I, 17 type II, 18 type III).

## Hsa-miR-9, -19a-3p, -23a-3p, -146a, and -206 were not differentially expressed in SMA serum samples

Some miRs were previously reported to be differentially expressed in serum samples of SMA patients (hsa-miR-9, -19a -3p, -23a -3p, -132, -146a , -183, -206, -431) (*Kye et al., 2014*; *Valsecchi et al., 2015*; *Murdocca et al., 2016*; *Catapano et al., 2016*; *Wertz et al., 2016*; *O'Hern et al., 2017*; *Sison et al., 2017*; *Kaifer et al., 2019*; *Bonanno et al., 2020*; *Haramati et al., 2010*). We evaluated whether these miRs were differentially expressed also in our samples (results are schematized in *Table 1*). miR-9, -132, -183, -206, and -431 were not differentially expressed in muscle samples, whereas miR-19a-3p and -146a were upregulated; miR-23a-3p was downregulated. miR-19a-3p, -146a , and -23a -3p were

upregulated in serum samples of patients when evaluated by r-qPCR; these preliminary data were not confirmed by a-qPCR. Also miR-206 and -9 were not differentially expressed by a-qPCR. At the time of study design, miR-132, -183, and -431 were not identified yet and thus have not been tested.

## The SMA-score: phenotypic severity can be predicted by combining *SMN2* copy number, *SMN2*-fl, miR-181a-5p, miR-324-5p, miR-451a, and age

We evaluated whether SMA-miR levels in serum could improve the accuracy of phenotype prediction with respect to the available molecular biomarkers. *SMN2* copy number determination alone provided a moderate accuracy when related to SMA type ($R^2$ = 52.45%, n = 41, p<$10^{-5}$; *Figure 5—figure supplement 1A*); the use of the decimal classification of SMA raised the $R^2$ up to 67.04% (n = 39, p<$10^{-5}$; *Figure 5—figure supplement 1B*). None of the patients had the rs121909192 or rs1454173648 variants. Considering also age at sampling in a multiple regression model, $R^2$ raised to 61.58%  ($R^2$ = 53.16%, n = 22, p=0.0005, in patients ≤ 6 years). When including also *SMN2*-fl levels and the sum of SMA-miRs as covariates, $R^2$ further raised up to 67.04%  (n = 40, p<$10^{-5}$); more importantly, when considering only patients ≤ 6 years, $R^2$ raised to 72.17%   (n = 21, p=0.0001). The equations describing the multivariate models were as follows.

> All ages:
> SMA type = 0.2473 + 0.0013 * age (months) + 0.0013 * *SMN2*-fl + 0.5417 * #SMN2 +0.00002 * SMA-miRs.
> Age ≤6 years:
> SMA type = –0.0700 + 0.0175 * age (months) + 0.0021 * *SMN2*-fl + 0.4398 * #SMN2 +0.00001 * SMA-miRs.

We compared by linear regression models the SMA-scores obtained with the two equations above in patients ≤ 6 years; the correlation coefficient was 0.90, and $R^2$ 80.31 (p<$10^{-5}$, n = 21, *Figure 5—figure supplement 1D*).

Then, we related the SMA subtype for each patient as from the equations above, with the decimal classification obtained from the blind evaluation. We found a correlation coefficient of 0.87 ($R^2$ = 75.77%, n = 38, p<$10^{-5}$) for the whole group and 0.87 ($R^2$ = 77.14%, n = 21, p<$10^{-5}$) for patients ≤ 6 years (*Figure 5A and B*). When evaluating patients with three *SMN2* copies only, the decimal classification and the SMA-score were significantly related ($R^2$ = 30.04, p=0.008, n = 21, *Figure 5—figure supplement 1C*).

## Discussion

The landscape of SMA has been so revolutionized over the last few years by the availability of effective treatments that the solution of past issues has become more urgent and novel ones have come to the surface. Firstly, the usual clinical classification is unsatisfactory for several reasons: (1) the treatment of patients has uncovered novel phenotypes that do not fall in any of the classical forms (*Mercuri et al., 2020*); (2) the spreading of newborn screening programs is changing the diagnosis of SMA into that of subjects with a genetic defect who might or not develop signs of the condition. Secondly, the identification of prognostic response and predictive biomarkers has become even more urgent since (1) the available outcome measures may not be sensitive enough to detect slight improvements that may still be clinically relevant and (2) the molecular effect of treatments that target CNS only cannot be evaluated peripherally.

The only genomic biomarker with clinical relevance is the determination of *SMN2* copy number, alongside two alternative splicing-modulating variants (rs121909192 and rs1454173648; *Vezain et al., 2010*; *Wu et al., 2017*) even though some other modifier genes have been reported (*Kariyawasam et al., 2019*). In this study, we have exploited an unbiased approach to identify deregulated muscular miRs that could be dosed in serum samples as candidate biomarkers for SMA; at the same time, high-throughput data might provide hints on the possible pathogenic role of skeletal muscle. miRs and neurodegeneration processes are tightly related: the depletion of DICER in mice leads to a phenotype resembling SMA (*Gonçalves et al., 2018*); DROSHA is downregulated in motor neurons of a SMA model (*Haramati et al., 2010*). In this latter study, the depletion of the bouquet of expressed miRs was interpreted as due to a global deregulation of miR biogenesis. In our study, less than 10%

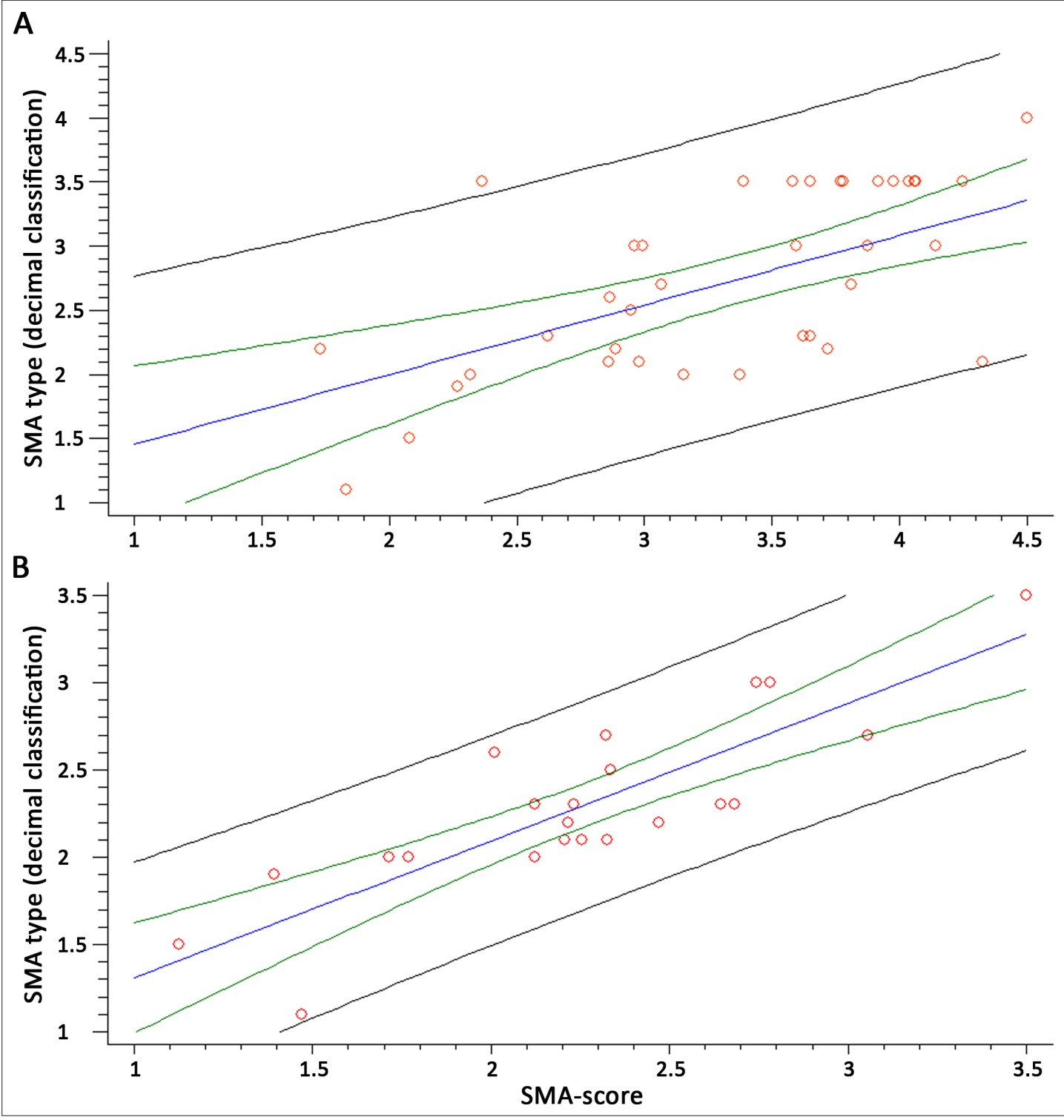

**Figure 5.** The spinal muscular atrophy score (SMA-score) predicts the phenotypic severity in SMA patients. Correlation between the SMA-score and the clinical decimal SMA subtype in the whole cohort (**A**) and aged ≤6 years (**B**). Red circles are individual samples, the blue line indicates the expected distribution, the green line indicates the 95% confidence interval, and the black lines are the prediction interval.

The online version of this article includes the following figure supplement(s) for figure 5:

**Figure supplement 1.** Linear correlation analysis among *SMN2* copy number and spinal muscular atrophy (SMA) type, estimated by the standard classification (**A**; $R^2$ = 52.45%, n = 41, $p<10^{-5}$) and the decimal classification (**B**; $R^2$ = 67.04%, n = 39, $p<10^{-5}$).

of miRs (100/1115 annotated miRs) were significantly deregulated in patients' muscle biopsies. This discrepancy might be related to the obvious difference in the tissues analyzed. More importantly, while we used human tissues, the motor neurons studied by Goncalves et al. were from the Taiwanese murine model of SMA that displays very low SMN levels, incomparable with those found in patients.

The involvement of skeletal muscle in SMA is unquestionable; to discern whether active or passive, we used the dual approach of muscle cell cultures and biopsies. While alterations found in vivo only could have been either primitive or secondary to the denervation/atrophy processes, the alterations found also in cultured cells point to a primary defect related to SMN deficiency.

In cell cultures, the number of differentially expressed miRs was smaller than in muscle samples. This finding could be related to the higher variability of cultured cells compared to the in vivo specimens. Interestingly, of the five miRs shared in all groups of samples (hsa-miR-1, -133a , -133b, -204-5p , -208b all upregulated), four belonged to the myomiRs: miR-1 and miR-133 family levels inversely correlate with myogenic differentiation/proliferation status (*Kirby and McCarthy, 2013*); miR-208b overexpression favors proliferation to differentiation during skeletal muscle development (*Fu et al., 2020*). Globally, these data suggest a primary muscular defect in SMA, determining a delay in the differentiation path from myoblasts to maturely innervated myofibers. The maturation defect of SMA skeletal muscle has been historically reported in morphological studies ahead of the identification of *SMN1* (*Fidziańska et al., 1990*) and has been confirmed in more recent studies on human (*Martínez-Hernández et al., 2014*) and murine tissues (*Bricceno et al., 2014*). Additionally, Houzelle et al. have very recently shown that hsa-miR-204-5p  and -133b levels inversely correlate with the mitochondrial activity in skeletal muscle (*Houzelle et al., 2020*), suggesting that these miRs could be involved in the mitochondrial depletion observed in SMA (*Ripolone et al., 2015*). How miR modulation occurs is unknown: SMN is not directly binding miRs even though the SMN complex can assemble with these small RNAs (*Chen and Chen, 2019*); conversely, in silico predictions (*Dweep et al., 2014*) indicate that miR-133a, -133b, and -204-5p  bind the 3'-UTR of *SMN2*, and thus might downregulate SMN transcript/protein levels.

In this study, we used the unbiased dataset above to identify candidate SMN-independent biomarkers. We first tested >100 miRs in a small cohort of patients and controls to get the shortlist of potential biomarkers. Most miRs were not detectable in serum, indicating the integrity of the sarcolemma in SMA, differently from other neuromuscular conditions such as Duchenne muscular dystrophy (*Cacchiarelli et al., 2011*). Only three miRs were overexpressed in serum of patients (miR-181a-5p, miR-324-5p , and miR-451a, the SMA-miRs), suggesting that these might be actively secreted by skeletal muscle. We can hypothesize that these miRs, which do not modulate *SMN2*, may exert a loco-regional effect in muscle and have, at the same time, an at-distance-effect following secretion.

Regarding the possible pathophysiological function of the SMA-miRs, to our knowledge, few data are available on the role of miR-324-5p  in skeletal muscle: a single study reported the upregulation in human CD56+ myoblasts, during differentiation (*Dmitriev et al., 2013*), again supporting the immaturity of SMA skeletal muscle. The putative role of the secreted miR-324-5p  remains unknown: *Sun et al., 2019* reported an essential function of this miR in synapse formation. Intriguingly, we observed that even if the intrathecal administration of the specific antagomir did not improve the survival of SMNΔ7 mice, however, it induced a significant transient increase in body weight (*Figure 4—figure supplement 1*).

More information is available regarding the role of miR-451a: an increase was found in aged muscle (*Mercken et al., 2013*), whereas the downregulation occurred during endurance training (*Zacharewicz et al., 2013*), suggesting that miR-451a levels may be inversely related to muscle mass. However, in our cohort, this miR was upregulated in patients independently of the degree of the atrophic process. Some studies have reported the simultaneous modulation of miR-451a and miR-181a-5p: in acute exercise both miRs are upregulated, whereas in aging these displayed an opposite trend (*Zacharewicz et al., 2013*), suggesting that the modulation in SMA is independent of the reduced mobility of patients; interestingly, we found that the expression levels of the two miRs are related (*Figure 3—figure supplement 1*). miR-181a-5p displays the more intriguing profile in terms of putative involvement in SMA: *Ouyang et al., 2012* reported a worsening of brain injury following the increase of miR levels in a mouse model of stroke, whereas the depletion accelerated the recovery. More importantly, *Benigni et al., 2016* found an increase in miR-181a levels in CSF of amyotrophic lateral sclerosis patients. We argue that miR-181a-5p might be a part of a retrograde signaling system from skeletal

muscle, which may accelerate motor neuron loss in SMA. To test this hypothesis, we treated SMNΔ7 mice with both mimic and anti-miR-181a-5p without increasing SMN levels. miR-181a-5p modulation alone was not sufficient to significantly improve the survival of affected mice even if some changes in Kaplan–Meier curves were observed in a subset of animals (*Figure 4*).

If confirmed in other studies with more therapeutic or pathogenic purposes, our data suggest that systemic therapeutic approaches increasing SMN levels also in skeletal muscle may provide additional benefits to SMA patients, and that miR-181a-5p (and/or miR324-5p) modulation might be a potential target for combinatorial treatments in addition to SMN modulation.

To the best of our knowledge, the miRs we have identified in the present study have not been described in SMA so far. Previously, other miRs have been found differentially expressed in animal models or in serum samples of patients (miR-9, -19a -3p, -23a -3p, -132, -146a , -183, -206, -431) (*Kye et al., 2014*; *Valsecchi et al., 2015*; *Murdocca et al., 2016*; *Catapano et al., 2016*; *Wertz et al., 2016*; *O'Hern et al., 2017*; *Sison et al., 2017*; *Kaifer et al., 2019*; *Bonanno et al., 2020*; *Haramati et al., 2010*). Of these (*Table 1*), three (miR-132, -183, -431) were not differentially expressed in muscle biopsies and had not been described at the time of study design, thus have not been tested here; among the others, three were almost undetectable in serum samples of our cohorts (miR-9, -19 a-3p, -146a ) and two were not differentially expressed (miR-23a-3p, -206). The discrepancy between our and previous data may be ascribed to the different technical approaches used: as in the case of *SMN2* mRNAs (*Tiziano et al., 2010*), the results of absolute qPCR, differently from the relative approaches, are not affected by the expression levels of endogenous genes and calibrators. Moreover, in the case of low-copy number transcripts, the relative approaches may magnify even small differences in expression levels, which may be statistically significant but of doubtful biological meaning.

One of the most relevant results of our study is the development of the SMA-score. The global spreading of newborn screening programs for SMA has made compelling the identification of tools with good predictive power of the clinical severity (*Dangouloff et al., 2020*). So far, the stop-or-go to the treatment of presymptomatic patients is uniquely based on *SMN2* copy number assessment, which is roughly predictive of the clinical severity in individual patients. Even more critical are patients with three *SMN2* copies whose severity may range from type I to type III (1.9 to 3b in our cohort).

Two *SMN2* variants (rs121909192 and rs1454173648) modulate the inclusion efficiency of exon 7 into mature mRNA, (*Vezain et al., 2010*; *Wu et al., 2017*), but these variants are relatively rare in patients. Additional *SMN2* variants have been very recently described (*Blasco-Pérez et al., 2021*); however, the frequency and the functional effect of these variants have not been elucidated yet. For these reasons, more functional and dynamic molecular markers could reasonably improve the prediction of the severity. The inclusion of serum SMA-miRs, whole-blood *SMN2*-fl levels, and age has markedly increased the accuracy of the severity prediction of *SMN2* copy number alone, from about 52% to about 75% . The age effect might be related to the physiological modulation of *SMN2*-fl levels over time, as previously reported (*Crawford et al., 2012*). In patients with three *SMN2* copies, the SMA-score was significantly related to the decimal classification of patients, even if with about 30% strength (*Figure 5—figure supplement 1C*). The increase in the population size might improve these results.

The main drawback of our results is related to the cross-sectional nature of the present study. While we have previously shown that *SMN2*-fl levels are stable in untreated patients for >1 year (*Tiziano et al., 2019*), longitudinal data on SMA-miR stability are lacking. In any case, repeated samplings could not be feasible in our study since almost all patients (except for the few type I subjects) were treated with SMN-modifying compounds (such as salbutamol; *Tiziano et al., 2019*; *Tiziano et al., 2010*; *Angelozzi et al., 2008*) or new experimental treatments. Also, the effect of SMN-modifying treatments on SMA-miRs is unknown. The collection of longitudinal data would be highly desirable, namely in presymptomatic patients, identified in our and other newborn screening projects (*Dangouloff et al., 2020*). If the accuracy will be confirmed in replicative studies, the SMA-score might be included in the clinical routine, as part of the prognostic process, once newborn testing will be universally available.

## Acknowledgements

We are very grateful to Famiglie SMA and to single patients and families for the continuous support. This study was granted by Fondazione Telethon Italia (#GGP12116) to FDT and LDM. CC was supported

by Girotondo/ONLUS. LLS was supported by a FIRC-AIRC fellowship for Italy. LLP was supported by ELIXIR IIB (https://elixir-europe.org/), the Italian Node of ELIXIR – the European Research Infrastructure for Life Science Data.

## Additional information

### Competing interests
Davide D'Amico: Davide D'Amico is affiliated with Amazentis SA. The author has no financial interests to declare. At the time of study developement, Dr. D'Amico had an academic affiliation. The other authors declare that no competing interests exist.

### Funding

| Funder | Grant reference number | Author |
|---|---|---|
| Fondazione Telethon | GGP12116 | Francesco Danilo Tiziano Lucia Di Marcotullio |

The funders had no role in study design, data collection and interpretation, or the decision to submit the work for publication.

### Author contributions
Emanuela Abiusi, Data curation, Investigation, Writing - original draft; Paola Infante, Investigation, Methodology, Data curation; Cinzia Cagnoli, Investigation, Methodology, Writing – review and editing; Ludovica Lospinoso Severini, Investigation, Methodology; Marika Pane, Lucia Ovidia Morandi, Eugenio Mercuri, Investigation, Supervision; Giorgia Coratti, Maria Carmela Pera, Adele D'Amico, Federica Diano, Agnese Novelli, Stefania Fiori, Giovanni Baranello, Isabella Moroni, Marina Mora, Maria Barbara Pasanisi, Krizia Pocino, Lorena Travaglini, Francesco Ria, Claudio Bruno, Denise Locatelli, Enrico Silvio Bertini, Investigation; Serena Spartano, Loredana Le Pera, Davide D'Amico, Investigation, Software; Lucia Di Marcotullio, Project administration, Supervision, Writing – review and editing, Conceptualization, Data curation, Funding acquisition; Francesco Danilo Tiziano, Conceptualization, Data curation, Formal analysis, Funding acquisition, Methodology, Project administration, Supervision, Writing – review and editing

### Author ORCIDs
Emanuela Abiusi https://orcid.org/0000-0001-9028-012X
Paola Infante https://orcid.org/0000-0003-0682-3916
Cinzia Cagnoli https://orcid.org/0000-0001-6863-6687
Ludovica Lospinoso Severini https://orcid.org/0000-0002-1032-5496
Loredana Le Pera https://orcid.org/0000-0002-0076-9878
Lucia Di Marcotullio https://orcid.org/0000-0003-0274-7178
Francesco Danilo Tiziano https://orcid.org/0000-0002-5545-6158

### Ethics
Human subjects: Informed consent was obtained from patients for genetic analyses. The study was approved by the local Ethics Committee.
According to the ARRIVE guidelines, procedures were carried out to minimize discomfort and pain, in compliance with National (D.L. 116 Suppl 40/1992 and D.L. 26/2014) and International guidelines and laws (2010/63/EU Legislation for the protection of animals used for scientific purposes). The experimental protocols were approved by the Ethics Committee of the Fondazione IRCCS Istituto Neurologico C. Besta and by the Italian Ministry of Health (protocol numbers: 962/2016-PR and 1039/2020-PR).

### Decision letter and Author response
Decision letter https://doi.org/10.7554/eLife.68054.sa1
Author response https://doi.org/10.7554/eLife.68054.sa2

# Additional files

## Supplementary files
• Supplementary file 1. Demographic and genetic characteristics of subjects who underwent muscle biopsy.
• Supplementary file 2. Clinical and molecular characteristics of subjects included in the present study.
• Supplementary file 3. Primer sequences.
• Supplementary file 4. List of deregulated miRs in whole miRNome analyses (patients vs. controls).
• Supplementary file 5. Assessment by relative and/or absolute qPCR of miR levels in serum samples of patients and controls.
• Transparent reporting form

## Data availability
All data generated or analysed during this study are included in the manuscript and supporting files. Raw sequencing data are available at NCBI-SRA database; BioProject PRJNA748014.

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
