## [Decision Letter]

**Acceptance summary:**

The most interesting aspects of the study identification of potential biomarkers that could be applied to treatment decisions at a time when therapies for SMA are now available. If confirmed, the findings would be of broad interest for therapeutics and for greater understanding of the role of miRs in SMA pathogenesis.

**Decision letter after peer review:**

Thank you for submitting your article "SMA-miRs (miR-181a-5p, -324-5p, and -451a) are overexpressed in spinal muscular atrophy skeletal muscle and serum samples" for consideration by *eLife*. Your article has been reviewed by 3 peer reviewers, including Christopher Cardozo as Reviewing Editor and Reviewer #1, and the evaluation has been overseen by Mone Zaidi as the Senior Editor. The following individual involved in review of your submission has agreed to reveal their identity: Zachary Graham (Reviewer #2).

Essential revisions:

General comments:

1. Data that is listed as 'available on request' or 'data not shown', should be provided in a Supplementary file either with the journal or a 3rd party repository (figshare or similar). In this age of almost unlimited file space, all data reported should be directly presented to the reader.

2. Figures should be redone with open boxes so that all of the data points can be easily seen.

3. This Italian cohort of authors should be commended for their preparation of the manuscript in English. However, there are many sections (such as paragraph 4 of the introduction) that would be greatly improved by a careful proofread by a native English speaker.

4. One abbreviation of microRNA, either miR or miRNA, should be used throughout the paper.

5. Background information on what miRNA are and their biological affect would be useful.

6. It is unclear what 51 serum/RNA/DNA samples means. Does it simply mean the authors collected 51 samples, then isolated RNA and DNA from it? Or were samples received with both RNA and DNA already isolated?

7. Exact p values should be listed uniformly throughout the paper (some figures have asterisks or section have p<0.05).

Specific Comments

Abstract

8. Under results, add that the SMA-score improved prediction to > 80% for those under age 6.

9. Under Discussion – it seems premature to speculate that the role of the differentially altered miRs was in programs of satellite cell differentiation. Keep in mind that the muscle biopsies sampled primarily muscle fibers and that satellite cells in those biopsies were quiescent, rather than the active proliferating form they take as myoblasts. Is it not also possible that these miRs target other key genes in mature myofibers? In addition, the paper really did not delve into role(s) of SMA in muscle per se, and did not try to link SMA1/2 levels to expression of the miRs identified. Thus, to me at least, any comment about function of SMA in muscle seems inappropriate for the abstract for this paper.

Introduction

10. The authors do not describe what SMN1/2 encodes, what its biological function is, and why genetic disruption of its expression is so detrimental to muscle function. A brief paragraph providing some general background on SMN would be appreciated for readers not familiar with SMA. Additional notes of onset, severity and phenotype differences between SMA I-III would also be useful to provide some additional background for the disease.

11. Paragraph #2 of the introduction should be removed or worked into other portions of the paper.

Methods

12. If possible, a more clear definition of what the authors mean by 'first stages' using, for example, clinical staging criteria, would improve the manuscript.

13. What passage numbers were use for cell cultures? How were the primary myoblasts isolated and stored (i.e. was CD56+ screening used)? These are important for people that may wish to try to replicate your study.

14. Please list the exact local ethics committee (hospital or university IRB or equivalent) and approval number.

15. Under 'Samples', add a reason that those controls were selected for the studies. Under discussion, review whether selection of these controls could be a confound.

16. Please provide a method for isolating/freezing myoblasts or cite your prior work if you have published these techniques.

17. Under 'Patients', were the authors involved in assigning SMA types blinded to the patient's diagnosis?

18. Under 'Whole miRome Sequencing', what filters were used (e.g., exclude miRs under 100 counts, etc). Are the raw read data available? Were they deposited in NIH GEO or other databases?

19. Under 'Statistical Tests' please clarify if an FDR threshold was applied for non-parametric tests (Mann-Whitney U-test). Also, please expand on how the multivariate analysis was performed in sufficient detail that others could use the sama analysis technique.

Results and Discussion

20. p values reported for miR-324 and 451 in Results section for Figure 3 do not match those listed on the figures for Figure 3 (.02 and.004 in Results section,.05 and.001 on the figures).

21. A significant weakness of the manuscript is that the experiments with injection of miRs into ventricles of the brain did not include controls to confirm that the miRs influenced mRNA or protein levels in the central nervous system. The most logical place to look for this disease of lower motor neurons would be spinal cord. Also unclear is why the miR were delivered to the central nervous system rather than skeletal muscle where they arose. Conceptually, any effects of the miRs identified in muscle and the circulation would be most likely to be evident in muscle or, possibly, at the nerve terminal; alternatively, circulating miRs could influence gene expression at any location in the body. I do not know if miRs cross the blood brain barrier.

22. It is quite difficult to figure out whether muscle or serum levels of miRs were used in the multivariable regression. The text of the results and discussion should be revised accordingly.

23. Relative abundance is one key parameter to consider when evaluating biological significance of miR expression. A comment regarding which of the differentially expressed miRs are highly expressed would be appropriate. The myoMirs are generally highly expressed in muscle and would be expected to be included among the highly expressed miRs in this dataset.

*Reviewer #1 (Recommendations for the authors):*

In this manuscript, the authors aimed to define a set of micro-RNAs (miRs) that were differentially expressed in patients with spinal muscular atrophy (SMA), to determine whether any of these miRs had roles in the disease, to validate prior data on altered miR expression in muscle of SMA patients with mutations of SMA 1, and to use information on miR levels in serum as a biomarker to improve the accuracy with which disease severity was predicted.

Strengths of the study include examination of miRs as a potential disease modifier in muscle biopsy samples and serum from a relatively large number of participants studied for relatively rare condition, properly controlled analysis to identify differentially expressed miRs, well controlled cell cultures studie of effects of those miRs on SMA 1 expression and an interesting algorithm for increasing disease severity prediction.

Experiments testing effects of injection into cerebral ventricles showed no effect of miRs on survival of SMA δ-7 mice, but did not report levels of these miRs in spinal cord or effects of the injection on known targets in spinal cord tissues thus limiting interpretation of the data.

If independently validated using a separate cohort, these findings may improve counseling and personalized treatment of patients with SMA.

The manuscript would be considerably improved if the following points were addressed.

General comments:

The authors refer to multiple sets of data that has not been included in the manuscript. These data should be added as supplemental material.

Figures should be redone with open boxes so that all of the data points can be easily seen.

The manuscript would benefit from careful editing for English, typographical errors and punctuation.

Overall, while the data from the manuscript are very interesting, it is a bit unfocused and includes experiments that address several different questions such as which miRs are differentially expressed in muscle, whether these changes are observed in serum, whether changing expression of these miRs in the nervous system influences disease progression, and whether circulating miRs could be a clinically useful biomarker for disease severity.

Specific Comments

Abstract

Under results, add that the SMA-score improved prediction to > 80% for those under age 6.

Under Discussion – it seems premature to speculate that the role of the differentially altered miRs was in programs of satellite cell differentiation. Keep in mind that the muscle biopsies sampled primarily muscle fibers and that satellite cells in those biopsies were quiescent, rather than the active proliferating form they take as myoblasts. Is it not also possible that these miRs target other key genes in mature myofibers? In addition, the paper really did not delve into role(s) of SMA in muscle per se, and did not try to link SMA1/2 levels to expression of the miRs identified. Thus, to me at least, any comment about function of SMA in muscle seems inappropriate for the abstract for this paper.

Methods

Under 'Samples', add a reason that those controls were selected for the studies. Under discussion, review whether selection of these controls could be a confound.

Please provide a method for isolating/freezing myoblasts or cite your prior work if you have published these techniques.

Under 'Patients', were the authors involved in assigning SMA types blinded to the patient's diagnosis?

Under 'Whole miRome Sequencing', what filters were used (e.g., exclude miRs under 100 counts, etc). Are the raw read data available? Were they deposited in NIH GEO or other databases?

Under 'Statistical Tests' please clarify if an FDR threshold was applied for non-parametric tests (Mann-Whitney U-test).

Results and Discussion

A significant weakness of the manuscript is that the experiments with injection of miRs into ventricles of the brain did not include controls to confirm that the miRs influenced mRNA or protein levels in the central nervous system. The most logical place to look for this disease of lower motor neurons would be spinal cord. Also unclear is why the miR were delivered to the central nervous system rather than skeletal muscle where they arose. Conceptually, any effects of the miRs identified in muscle and the circulation would be most likely to be evident in muscle or, possibly, at the nerve terminal; alternatively, circulating miRs could influence gene expression at any location in the body. I do not know if miRs cross the blood brain barrier.

It is quite difficult to figure out whether muscle or serum levels of miRs were used in the multivariable regression. The text of the results and discussion should be revised accordingly.

Relative abundance is one key parameter to consider when evaluating biological significance of miR expression. A comment regarding which of the differentially expressed miRs are highly expressed would be appropriate. The myoMirs are generally highly expressed in muscle and would be expected to be included among the highly expressed miRs in this dataset.

*Reviewer #2 (Recommendations for the authors):*

Abiusi et al., describe a novel pipeline for predicting severity of infantile spinal muscular atrophy (SMA). They use muscle biopsies and primary cell cultures as well as serum from individuals with SMA. For additional studies, they use an SMA mouse model with miRNA anti-sense injections. They conclude their new model was able to improve prediction of phenotypic severity when related to patient subtype and stratified by age. While the paper has a solid base, I think there are areas that need to be clarified, strengthened or carefully considered. I thank the authors for doing this work and putting together the manuscript. I hope the following will help solidify and improve aspects of the paper.

Strengths

– SMA is a deleterious genetic disease and the ability to improve prediction of severity would be an important clinical outcome.

– Use of miRNA sequencing to find biomarkers is a solid experimental approach.

– Coupling biomarkers with genetic outcomes is a logical predictive strategy.

– Follow-up testing on previously proposed biomarkers.

Weaknesses

– There is a disparity in FDRs used for various miRNA analyses. Reasons for differences in the accepted FDR should be explained and also may have affected downstream analyses. FDRs listed for miRNome was said to be selected at p<0.01 in the methods, with muscle having 99, myoblasts having 20 and myotubes having 19 DE miRNA. However, using their stated FDR of p<0.01, two of their selected predictors, miR-181 and miR-324 do not meet that threshold for being differentially expressed. Then there is a switch to accepting an FDR of p<0.05. Uniformity of data analyses is crucial for biomarker studies and this is a major oversight.

– For the first tier round of qPCR follow up, how and why where those individuals selected out of the complete cohort? Having 1 SMA1 and 9 SMA II seems like a very oddly weighted group breakdown.

– How was type I error limited for the absolute qPCR? FDRs of relative qPCR were stated to be an FDR but nothing is mentioned for absolute qPCR. This is important considering 50 assays were completed for these outcomes.

– A table showing all the outcomes of the relative and absolute qPCR would provide a great deal of information and transparency. For example, the miRNA selected as 'predictors' for SMA were upregulated, but biomarkers can also be down-regulated.

– The link of SMN being depleted in muscle having primary effects is intriguing, but should be interpreted with caution. The importance of having a tonic motor connection with a healthy motor neuron cannot be underestimated. Changes in differentiation factors in proliferating myoblasts and forming myotubes related to myomiR are often seen in other pathological conditions that result from denervation (such as Parkinson's, age-related denervation and ALS). Additionally, only these factors were considered and not any phenotypic measurements of differentiating myotubes. The data are intriguing, but directed and focused studies are needed to investigate how muscle may or may not be regulated by global SMN1 reductions.

– There is no description of when the biopsies of the SMA patients were taken. As they say they were taken from the quadriceps during the first stages of the disease, this could be quite variable depending on fiber type of the muscle group the biopsy was taken from, time diagnosis was confirmed and the age at the time of biopsy. Clarification of these points, as well as for the primary tissue cultures, is important.

– It is not listed where the serum from individuals with SMA came from. Please explain if these were obtained from a repository similar to the muscle tissues or if separate process was conducted.

– The disparity in group sizes from type I to type II and III is unfortunate. It is understood the difference in disease disparity causes this, but it is a limitation in the search for prognostic biomarkers to have such unbalanced group differences.

– Assignment of patients to the SMA subgroup is vague and likely difficult to understand for non-clinicians. Explicit descriptions of how each individual used the clinical data available to stratify patients would be appreciated.

– The authors do not describe what SMN1/2 encodes, what its biological function is, and why genetic disruption of its expression is so detrimental to muscle function. A brief paragraph providing some general background on SMN would be appreciated for readers not familiar with SMA. Additional notes of onset, severity and phenotype differences between SMA I-III would also be useful to provide some additional background for the disease.

– Data that is listed as 'available on request' or 'data not shown', should be provided in a Supplementary file either with the journal or 3rd party repository (figshare or similar). In this age of almost unlimited file space, all data reported should be directly presented to the reader.

– Paragraph #2 of the introduction should be removed or worked into other portions of the paper.

– This Italian cohort of authors should be commended for their preparation of the manuscript in English. However, there are many sections (such as paragraph 4 of the introduction) that would be greatly improved by a careful proofread by a native English speaker.

– One abbreviation of microRNA, either miR or miRNA, should be used throughout the paper.

– Background information on what miRNA are and their biological affect would be useful.

– It is unclear what 51 serum/RNA/DNA samples means. Does it simply mean the authors collected 51 samples, then isolated RNA and DNA from it? Or were samples received with both RNA and DNA already isolated?

– Exact p values should be listed uniformly throughout the paper (some figures have asterisks or section have p<0.05).

– What passage numbers were use for cell cultures? How were the primary myoblasts isolated and stored (i.e. was CD56+ screening used)? These are important for people that may wish to try to replicate your study.

– Please list the exact local ethics committee (hospital or university IRB or equivalent) and approval number.

– p values reported for miR-324 and 451 in Results section for Figure 3 do not match those listed on the figures for Figure 3 (.02 and.004 in Results section,.05 and.001 on the figures).

–Related to figures, improved opacity to allow the reader to see the individual data points would be appreciated.

*Reviewer #3 (Recommendations for the authors):*

The manuscript by Abiusi E, Infante P and co-workers describes a study to discover and validate miRNAs in muscle of mice models of SMA and biopsies and cultures of SMA patients. Furthermore, the authors study their presence in serum of 51 patients and report an attempt to correlate results in serum miRNAs with SMN2 copy number, FL transcripts and age of the SMA patients under study. The resulting correlation is named SMA-score.

The authors managed to develop an appropriate streamline to detect and validate candidate miRNAs. Data of specific miRNAs is interesting but at present there is lack of longitudinal analysis or trajectories to ascribe them a value as dynamic biomarkers. These issues should be investigated to ascribe a potential role of miRNAs as biomarkers and to further validate the utility of the SMA-score.

SMN2 analysis of these patients was appropriate including the variants rs121909192 and rs1454173648 as positive modifiers (that were discarded in the cases under study) strengthen the accuracy of this variable to include into the score. Further investigation of the entire sequence of the SMN2 of patients may provide additional information about prediction of severity and would be a next step to help improving accuracy of SMA prediction and possibly will be included in the score.

1) Title. I would edit the title to reflect a more realistic scenario of this investigation that is the discovering and validation of miRNAs. The application of the SMA-score is in my opinion preliminary (see below further comments).

2) Abstract. Edits are also necessary in the abstract. i.e. there is incomplete information given that a part of these studies were performed in mice models; wording: "evaluate its controversial pathogenic role" is not addressed in this work and the conclusion that "may provide additional benefit to SMA patients" is not new; also "the SMA-score could be crucial for the prognostic assessment of pre-symptomatic patients" is rather speculative given the lack of data in pre-symptomatic patients regarding the candidates miRNAs and the score application. In summary, the abstract should be more focused in the issues of the study.

3) Methods. The authors should clarify the meaning of "first stages", does mean that all the samples were taken at the beginning of manifestations? Perhaps a Table (could be also supplementary) can be included with a summary of the samples studied with additional information of first stages.

4) Discovering and data of specific miRNAs is interesting and the analysis is adequate. The lack of longitudinal analysis or trajectories to ascribe them a value as dynamic biomarkers should be commented.

5) The cohort of type I patients is small in my opinion to establish a correlation (only four patients). But it could be interesting to try a correlation between the values of type II versus type III cases to determine if patients with three copies acquire (type III) or not (type II) the ability for autonomous walking.

6) There are four patients in the list of type II and III disease with two SMN2 copies. In absence of positive modifier variants (rs121909192 and rs1454173648 also known as NM_017411.3:c.859G>C and NM_017411.3:c.835-44A>G)(both nomenclatures can be included for clarity in the paper) these "outliers" may influence the final score? i.e. PT11 2.70 vs. PT 38 2.12 >6 years (according to data set). Values should be with period not commas, the same applies to the equation.

7) What explanation is feasible to explain the correlation differences in > or < 6 years? Can the authors clarify scores >6 years or < 6 years from the same patient?

8) Discussion: neurofilament should be referenced (i.e. Darras BT, et al., Ann Clin Transl Neurol. 2019 Apr 17;6(5):932-944. doi: 10.1002/acn3.779. PMID: 31139691; PMCID: PMC6530526.

9) The authors provide additional evidence that muscle is involved in SMA but as they said in Discussion, there is some heterogeneity in the samples. The statement "Taken together, these data suggest that systemic therapeutic approaches increasing SMN levels also in skeletal muscle tissues may provide additional benefits to SMA patients" remains speculative according to what is discussed below this statement about miR-324-5p, miR-451a and miR-181a-5p. I would try to elaborate a statement after the discussion of the role in muscle of the miRNAs detected with a more realistic conclusion.

10) "More information is (are) available".

11) Please clarify some aspects of the multivariate analysis performed. Selection of variables that are introduced in the multivariate analysis can be done by a full model (the more variables you enter, the better R2 you will have…) or selection of only those variables with significant value in the bivariate linear regression (one by one). The authors state: "Continuous variables were compared by linear correlations. Multivariate analysis was used to correlate severity and molecular parameters (miRNA levels, SMN2 transcripts, SMN2 copy number)". Which variables were significant if compared one by one?

12) Please clarify in Table 1 the meaning of P in the column of absolute qPCR.

13) The statement "The inclusion of SMA-miRs, age and SMN2-fl levels has almost doubled the accuracy of the severity prediction of SMN2 copy number alone, from about 43% to 82%." should be better discussed. Data of calculations on SMN2 alone (according to the authors apparently is 43%) should be provided because is confusing. Looking at the data set Table, in type III there are 13 with 4 copies (54%), 9 with 3 copies (37%) and 2 with 2 copies (8%). In a total of 18 type II patients the score was calculated and 15 (83%) had 3 SMN2 copies as expected in these type of patients. Adding the 4 type I patients the total number of patients with calculated score is 46 (just serum was available in 51 but the score was performed in 46, please mention in the text).

14) I found 7/24 (30%) cases with type III disease with an insufficient SMA-score (less than 2.5 assuming in type II disease according to the score). It is possible to discuss these results?

15) As the authors mentioned in the last paragraphs, data in newborns (control and SMA) and trajectories with longitudinal data and evolution of the miRNAS should be investigated to ascribe a potential role as a biomarker and to validate the utility of the score.

16) Even though the variants rs121909192 and rs1454173648 were discarded, a sentence may be included mentioning that investigation of the entire sequence of the SMN2 of patients (perhaps with a reference) may provide additional information about prediction of severity and would be a next step to help improving accuracy of prediction.

---

## [Author Response]

Essential revisions:General comments:1. Data that is listed as 'available on request' or 'data not shown', should be provided in a Supplementary file either with the journal or a 3rd party repository (figshare or similar). In this age of almost unlimited file space, all data reported should be directly presented to the reader.

Data previously indicated as “not shown” have been included in the manuscript, mainly as supplementary information (see supplementary figures 1-6 and tables).

2. Figures should be redone with open boxes so that all of the data points can be easily seen.

As requested, figures have been re-drawn. In order to better appreciate the whole distribution of miRNA levels in patients and controls, we have changed the Y-axis scale of figure 3 from linear to exponential. The same approach has been used for the novel supplementary figures 2 and 3.

3. This Italian cohort of authors should be commended for their preparation of the manuscript in English. However, there are many sections (such as paragraph 4 of the introduction) that would be greatly improved by a careful proofread by a native English speaker.

We have extensively revised the manuscript both for the form and possible typos.

4. One abbreviation of microRNA, either miR or miRNA, should be used throughout the paper.

As requested, we opted for miR as a unique abbreviation for miRNA.

5. Background information on what miRNA are and their biological affect would be useful.

As requested, we added some sentences and references regarding this matter.

6. It is unclear what 51 serum/RNA/DNA samples means. Does it simply mean the authors collected 51 samples, then isolated RNA and DNA from it? Or were samples received with both RNA and DNA already isolated?

We have specified the number of each specimen available in the Method section.

7. Exact p values should be listed uniformly throughout the paper (some figures have asterisks or section have p<0.05).

We have uniformly listed the p values throughout the paper.

Specific CommentsAbstract8. Under results, add that the SMA-score improved prediction to > 80% for those under age 6.9. Under Discussion – it seems premature to speculate that the role of the differentially altered miRs was in programs of satellite cell differentiation. Keep in mind that the muscle biopsies sampled primarily muscle fibers and that satellite cells in those biopsies were quiescent, rather than the active proliferating form they take as myoblasts. Is it not also possible that these miRs target other key genes in mature myofibers? In addition, the paper really did not delve into role(s) of SMA in muscle per se, and did not try to link SMA1/2 levels to expression of the miRs identified. Thus, to me at least, any comment about function of SMA in muscle seems inappropriate for the abstract for this paper.

The abstract has been almost completely rephrased.

Introduction10. The authors do not describe what SMN1/2 encodes, what its biological function is, and why genetic disruption of its expression is so detrimental to muscle function. A brief paragraph providing some general background on SMN would be appreciated for readers not familiar with SMA. Additional notes of onset, severity and phenotype differences between SMA I-III would also be useful to provide some additional background for the disease.

This point has been addressed in the Introduction section. We agree with the reviewer that further information on SMN function in muscle would be appreciable but, unfortunately, most detrimental mechanisms induced by SMN reduction are unknown, in particular in skeletal muscle but also in motor neurons.

11. Paragraph #2 of the introduction should be removed or worked into other portions of the paper.

This point has been addressed in the Introduction section.

Methods12. If possible, a more clear definition of what the authors mean by 'first stages' using, for example, clinical staging criteria, would improve the manuscript.

This point has been addressed, by adding Supplementary table 1. “First stages of disease” were defined as “as onset of the first clinical signs that prompted the diagnostic workflow”. Unfortunately, to the best of our knowledge, no clinical staging criteria are available, besides the usual classification of SMA.

13. What passage numbers were use for cell cultures? How were the primary myoblasts isolated and stored (i.e. was CD56+ screening used)? These are important for people that may wish to try to replicate your study.

Passages of cell cultures were between 5 and 15. For isolation of myoblasts, we used CD56 selection, as indicated in the ref Zanotti et al., 2007. These aspects were specified in the Methods.

14. Please list the exact local ethics committee (hospital or university IRB or equivalent) and approval number.

This information has been specified as requested in the Methods section.

15. Under 'Samples', add a reason that those controls were selected for the studies. Under discussion, review whether selection of these controls could be a confound.

This point has been addressed in the Methods section.

16. Please provide a method for isolating/freezing myoblasts or cite your prior work if you have published these techniques.

As reported above (point 13.) we have included our ref Zanotti et al., where experimental details were specified.

17. Under 'Patients', were the authors involved in assigning SMA types blinded to the patient's diagnosis?

Data on the clinical performance of patients, evaluated by functional scale scores and motor ability, are sufficient to assign a decimal SMA subtype, even if the gross classification (type I-III) is unknown. In this sense, the utility of molecular tests is very limited, due to the finding of the same genotype at the SMN locus, in patients with markedly different phenotypes.

18. Under 'Whole miRome Sequencing', what filters were used (e.g., exclude miRs under 100 counts, etc). Are the raw read data available? Were they deposited in NIH GEO or other databases?

The pipeline analysis has been specified in Methods. Raw data of miRSeq have been deposited at NCBI-SRA database (BioProject PRJNA748014). The significantly up- and down-regulated microRNAs were selected at False Discovery Rate (FDR) <0.05

19. Under 'Statistical Tests' please clarify if an FDR threshold was applied for non-parametric tests (Mann-Whitney U-test). Also, please expand on how the multivariate analysis was performed in sufficient detail that others could use the sama analysis technique.

Due to the analytical workflow we used for absolute qPCR, i.e. a progressive increase in the number of samples tested, in case of first tier significant results, we established an α-value at 0.05, without applying an FDR threshold. The rationale behind this strategy was that the number of miRNAs that were selected for the second tier absolute qPCR was small, thus minimizing the risk of false positive results, due to the number of variables. This aspect has been detailed in Methods section.

Results and Discussion20. p values reported for miR-324 and 451 in Results section for Figure 3 do not match those listed on the figures for Figure 3 (.02 and.004 in Results section,.05 and.001 on the figures).

Figure 3 has been replaced with a new version, as requested. Indeed, the proper p-values are those indicated in the text. Those indicated in the figures were originated in a previous version of the graphs. We apologize.

21. A significant weakness of the manuscript is that the experiments with injection of miRs into ventricles of the brain did not include controls to confirm that the miRs influenced mRNA or protein levels in the central nervous system. The most logical place to look for this disease of lower motor neurons would be spinal cord.

We agree with the reviewer that the putative mode of action of the SMA-miRs/antagomiRs has not been explored at all in our study. These aspects were, in our opinion, out of focus with respect to the main aim of our project that was essentially aimed at identifying SMN-independent biomarkers. For similar reasons, we did not check whether miR level modulation could affect SMN levels in CNS or namely in spinal cord. Additionally, two lines of evidences were in support of this strategy: (1) as shown in the present study, SMA-miRs do not modulate SMN levels in human cells of neuronal origin; (2) since the survival of treated mice is not affected by antagomiRs, we think unlikely a direct effect on SMN levels that, based on preclinical and clinical data, would have led to a positive effect on mice phenotype.

Also unclear is why the miR were delivered to the central nervous system rather than skeletal muscle where they arose. Conceptually, any effects of the miRs identified in muscle and the circulation would be most likely to be evident in muscle or, possibly, at the nerve terminal; alternatively, circulating miRs could influence gene expression at any location in the body. I do not know if miRs cross the blood brain barrier.

With our preliminary experiments (but more detailed studies on miR/antagomiR system are ongoing) we aimed at evaluating a specific hypothesis, that is the retrograde effect of SMA-miRs on CNS. We agree with the reviewer that it is conceivable that, based on the finding of secreted miRs in serum, these molecules are likely exert a systemic effect: to reduce possible confounding variables, we decided to restrict miR modulation to a specific target tissue, namely the CNS, rather than evaluating the loco-regional effect in skeletal muscle. However, several studies report the ability of extra-cellular vesicles to cross the blood brain barrier and to deliver their content to the CNS (see for example Izco et al., Neuroscientist 2021 Feb 3; 1073858421990001. doi: 10.1177/1073858421990001).

22. It is quite difficult to figure out whether muscle or serum levels of miRs were used in the multivariable regression. The text of the results and discussion should be revised accordingly.

This point has been addressed as requested

23. Relative abundance is one key parameter to consider when evaluating biological significance of miR expression. A comment regarding which of the differentially expressed miRs are highly expressed would be appropriate. The myoMirs are generally highly expressed in muscle and would be expected to be included among the highly expressed miRs in this dataset.

To address this point of the reviewer, we have included in Supplementary Table 4 also logCPM of miRNA levels in muscle samples, as obtained by NGS. Since miRNA levels in serum were determined by absolute qPCR, the relative abundance of the different miRs is reported as no. of molecules/μl of serum in figures and tables. As hypothesized by the reviewer, myo-miRs were abundantly expressed in muscle samples, at significantly higher levels in patients. Intriguingly, these miRNAs were not differentially expressed in serum, suggesting that, differently from the SMA-miRs, myo-miRs are not actively secreted.

Reviewer #1 (Recommendations for the authors):General comments:The authors refer to multiple sets of data that has not been included in the manuscript. These data should be added as supplemental material.

The following materials have been included as supplementary:

Supplementary file 1: demographic and clinical characteristics of patients who underwent to biopsy

Supplementary file 2: molecular and clinical details of the single patients

Supplementary file 4: detailed list of differential miRs, including the logCPM

Additionally, the raw data of NGS runs have been uploaded in a public repository (NCBI-SRA database, BioProject PRJNA748014)

Figures should be redone with open boxes so that all of the data points can be easily seen.

This point has been addressed as requested. In order to magnify the distribution of miR levels in our cohort, we have replaced the linear scale with an exponential.

The manuscript would benefit from careful editing for English, typographical errors and punctuation.

This point has been addressed as requested.

Overall, while the data from the manuscript are very interesting, it is a bit unfocused and includes experiments that address several different questions such as which miRs are differentially expressed in muscle, whether these changes are observed in serum, whether changing expression of these miRs in the nervous system influences disease progression, and whether circulating miRs could be a clinically useful biomarker for disease severity.Specific CommentsAbstractUnder results, add that the SMA-score improved prediction to > 80% for those under age 6.Under Discussion – it seems premature to speculate that the role of the differentially altered miRs was in programs of satellite cell differentiation. Keep in mind that the muscle biopsies sampled primarily muscle fibers and that satellite cells in those biopsies were quiescent, rather than the active proliferating form they take as myoblasts. Is it not also possible that these miRs target other key genes in mature myofibers? In addition, the paper really did not delve into role(s) of SMA in muscle per se, and did not try to link SMA1/2 levels to expression of the miRs identified. Thus, to me at least, any comment about function of SMA in muscle seems inappropriate for the abstract for this paper.

See the response to specific comments #9.

MethodsUnder 'Samples', add a reason that those controls were selected for the studies. Under discussion, review whether selection of these controls could be a confound.

This point has been addressed as requested in the Methods section.

Please provide a method for isolating/freezing myoblasts or cite your prior work if you have published these techniques.

See response to item 16 in the Specific comments.

Under 'Patients', were the authors involved in assigning SMA types blinded to the patient's diagnosis?

See response to item 17 in the Specific comments.

Under 'Whole miRome Sequencing', what filters were used (e.g., exclude miRs under 100 counts, etc). Are the raw read data available? Were they deposited in NIH GEO or other databases?Under 'Statistical Tests' please clarify if an FDR threshold was applied for non-parametric tests (Mann-Whitney U-test).

See response to item 19 in the Specific comments.

Results and DiscussionA significant weakness of the manuscript is that the experiments with injection of miRs into ventricles of the brain did not include controls to confirm that the miRs influenced mRNA or protein levels in the central nervous system. The most logical place to look for this disease of lower motor neurons would be spinal cord. Also unclear is why the miR were delivered to the central nervous system rather than skeletal muscle where they arose. Conceptually, any effects of the miRs identified in muscle and the circulation would be most likely to be evident in muscle or, possibly, at the nerve terminal; alternatively, circulating miRs could influence gene expression at any location in the body. I do not know if miRs cross the blood brain barrier.

See response to item 21 in the Specific comments.

It is quite difficult to figure out whether muscle or serum levels of miRs were used in the multivariable regression. The text of the results and discussion should be revised accordingly.

See response to item 22 in the Specific comments.

Relative abundance is one key parameter to consider when evaluating biological significance of miR expression. A comment regarding which of the differentially expressed miRs are highly expressed would be appropriate. The myoMirs are generally highly expressed in muscle and would be expected to be included among the highly expressed miRs in this dataset.

See response to item 23 in the Specific comments.

Reviewer #2 (Recommendations for the authors):Abiusi et al., describe a novel pipeline for predicting severity of infantile spinal muscular atrophy (SMA). They use muscle biopsies and primary cell cultures as well as serum from individuals with SMA. For additional studies, they use an SMA mouse model with miRNA anti-sense injections. They conclude their new model was able to improve prediction of phenotypic severity when related to patient subtype and stratified by age. While the paper has a solid base, I think there are areas that need to be clarified, strengthened or carefully considered. I thank the authors for doing this work and putting together the manuscript. I hope the following will help solidify and improve aspects of the paper.Strengths– SMA is a deleterious genetic disease and the ability to improve prediction of severity would be an important clinical outcome.– Use of miRNA sequencing to find biomarkers is a solid experimental approach.– Coupling biomarkers with genetic outcomes is a logical predictive strategy.– Follow-up testing on previously proposed biomarkers.Weaknesses– There is a disparity in FDRs used for various miRNA analyses. Reasons for differences in the accepted FDR should be explained and also may have affected downstream analyses. FDRs listed for miRNome was said to be selected at p<0.01 in the methods, with muscle having 99, myoblasts having 20 and myotubes having 19 DE miRNA. However, using their stated FDR of p<0.01, two of their selected predictors, miR-181 and miR-324 do not meet that threshold for being differentially expressed. Then there is a switch to accepting an FDR of p<0.05. Uniformity of data analyses is crucial for biomarker studies and this is a major oversight.

We are very grateful to the reviewer for noticing our mistake, due to a typo. The FDR for all analyses was selected <0.05. The text has been modified accordingly in the Methods section.

– For the first tier round of qPCR follow up, how and why where those individuals selected out of the complete cohort? Having 1 SMA1 and 9 SMA II seems like a very oddly weighted group breakdown.

Thank you very much for the constructive comment, we specified this information in the Results. At that stage, due to the lack of preliminary data, we opted for the more homogenous sub-cohort of patients, in terms of age and severity, to reduce potential biases in miR quantification. The type I patient was the only available at the time.

– How was type I error limited for the absolute qPCR? FDRs of relative qPCR were stated to be an FDR but nothing is mentioned for absolute qPCR. This is important considering 50 assays were completed for these outcomes.

See response to Specific comments #19.

– A table showing all the outcomes of the relative and absolute qPCR would provide a great deal of information and transparency. For example, the miRNA selected as 'predictors' for SMA were upregulated, but biomarkers can also be down-regulated.

As requested we have included a supplementary table (#5) reporting the results of the single miRNAs in relative and/or absolute qPCR.

– The link of SMN being depleted in muscle having primary effects is intriguing, but should be interpreted with caution. The importance of having a tonic motor connection with a healthy motor neuron cannot be underestimated. Changes in differentiation factors in proliferating myoblasts and forming myotubes related to myomiR are often seen in other pathological conditions that result from denervation (such as Parkinson's, age-related denervation and ALS). Additionally, only these factors were considered and not any phenotypic measurements of differentiating myotubes. The data are intriguing, but directed and focused studies are needed to investigate how muscle may or may not be regulated by global SMN1 reductions.

Thank you very much for the comment. We fully agree that our data alone do not allow drawing any conclusion on the pathogenic role of skeletal muscle in SMA, but this was not the primary aim of the study that has been designed for biomarker discovery. However, besides data obtained in murine models or those in human whole muscle samples, some in vitro studies on human muscle cell cultures have demonstrated primitive alterations of patients’ satellite cells, including the inability of SMA myotubes to sustain motor neuron innervation, when co-cultured with wild type motor neurons (see for example Braun et al., 1995; Guettier-Sigrist et al., 2002).

– There is no description of when the biopsies of the SMA patients were taken. As they say they were taken from the quadriceps during the first stages of the disease, this could be quite variable depending on fiber type of the muscle group the biopsy was taken from, time diagnosis was confirmed and the age at the time of biopsy. Clarification of these points, as well as for the primary tissue cultures, is important.

See response to item 12 in the Specific comments.

– It is not listed where the serum from individuals with SMA came from. Please explain if these were obtained from a repository similar to the muscle tissues or if separate process was conducted.

We agree with the reviewer. Patients were enrolled ad hoc in the participating neuromuscular centers. These patients are or were in routine clinical follow-up. This aspect has been specified in the Methods section.

– The disparity in group sizes from type I to type II and III is unfortunate. It is understood the difference in disease disparity causes this, but it is a limitation in the search for prognostic biomarkers to have such unbalanced group differences.

We fully agree with the reviewer. However, it is unlikely to succeed in balancing the sub-populations, in such wide phenotypic spectrum of disease, in terms of age of onset, severity and relative abundance of the different forms. Additionally, the total number of type I patients in our study has been limited also by the difficulties in sampling adequate amounts of blood from such young and/or hypotonic patients. Indeed, we had to exclude some other patients due to the lack of serum samples.

– Assignment of patients to the SMA subgroup is vague and likely difficult to understand for non-clinicians. Explicit descriptions of how each individual used the clinical data available to stratify patients would be appreciated.

In the Introduction, we specified some details on how SMA patients are classified, based on the clinical severity. Since this classification has been published several years ago, we preferred not to detail the single items used for scoring patients. The refs have been specified in Introduction and Methods section.

– The authors do not describe what SMN1/2 encodes, what its biological function is, and why genetic disruption of its expression is so detrimental to muscle function. A brief paragraph providing some general background on SMN would be appreciated for readers not familiar with SMA. Additional notes of onset, severity and phenotype differences between SMA I-III would also be useful to provide some additional background for the disease.

See response to Specific comments #10. As requested, we added some details on SMN.

– Data that is listed as 'available on request' or 'data not shown', should be provided in a Supplementary file either with the journal or 3rd party repository (figshare or similar). In this age of almost unlimited file space, all data reported should be directly presented to the reader.

See response to item 1 in the General Comments.

– Paragraph #2 of the introduction should be removed or worked into other portions of the paper.– This Italian cohort of authors should be commended for their preparation of the manuscript in English. However, there are many sections (such as paragraph 4 of the introduction) that would be greatly improved by a careful proofread by a native English speaker.

See response to item 3 in the General Comments.

– One abbreviation of microRNA, either miR or miRNA, should be used throughout the paper.

See response to item 4 in the General Comments.

– Background information on what miRNA are and their biological affect would be useful.

See response to item 5 in the General Comments.

– It is unclear what 51 serum/RNA/DNA samples means. Does it simply mean the authors collected 51 samples, then isolated RNA and DNA from it? Or were samples received with both RNA and DNA already isolated?

See response to item 6 in the General Comments.

– Exact p values should be listed uniformly throughout the paper (some figures have asterisks or section have p<0.05).

See response to item 7 in the General Comments.

– What passage numbers were use for cell cultures? How were the primary myoblasts isolated and stored (i.e. was CD56+ screening used)? These are important for people that may wish to try to replicate your study.

See response to item 13 in the General Comments.

– Please list the exact local ethics committee (hospital or university IRB or equivalent) and approval number.

See response to item 14 in the General Comments.

– p values reported for miR-324 and 451 in Results section for Figure 3 do not match those listed on the figures for Figure 3 (.02 and.004 in Results section,.05 and.001 on the figures).

See response to item 20 in the General Comments.

–Related to figures, improved opacity to allow the reader to see the individual data points would be appreciated.

See response to item 2 in the General Comments.

Reviewer #3 (Recommendations for the authors):1) Title. I would edit the title to reflect a more realistic scenario of this investigation that is the discovering and validation of miRNAs. The application of the SMA-score is in my opinion preliminary (see below further comments).

The title has been rephrased as requested.

2) Abstract. Edits are also necessary in the abstract. i.e. there is incomplete information given that a part of these studies were performed in mice models; wording: "evaluate its controversial pathogenic role" is not addressed in this work and the conclusion that "may provide additional benefit to SMA patients" is not new; also "the SMA-score could be crucial for the prognostic assessment of pre-symptomatic patients" is rather speculative given the lack of data in pre-symptomatic patients regarding the candidates miRNAs and the score application. In summary, the abstract should be more focused in the issues of the study.

The abstract has been rephrased as requested.

3) Methods. The authors should clarify the meaning of "first stages", does mean that all the samples were taken at the beginning of manifestations? Perhaps a Table (could be also supplementary) can be included with a summary of the samples studied with additional information of first stages.

As requested, a supplementary table has been added (#1). See response to item # 12 of the Specific comments.

4) Discovering and data of specific miRNAs is interesting and the analysis is adequate. The lack of longitudinal analysis or trajectories to ascribe them a value as dynamic biomarkers should be commented.

See response to item # 8 of the minor comments.

5) The cohort of type I patients is small in my opinion to establish a correlation (only four patients). But it could be interesting to try a correlation between the values of type II versus type III cases to determine if patients with three copies acquire (type III) or not (type II) the ability for autonomous walking.

We agree with the reviewer that the type I cohort is very small. However, due to the difficult sampling and to the amount of biological specimen required for the different analyses reported in this study, we had to exclude 4 additional patients due to the lack of serum or RNA samples. Additionally, the spreading of experimental treatments has complicated the enrollment of further naïve patients.

We re-evaluated with the referral clinicians the whole cohort and concluded that one of the patients has been erroneously attributed to type I while being type II. This aspect has been emended in table and figures, and the equations have been concordantly corrected. These modifications have led to a further improvement of the correlation between the SMA-score and the SMA decimal classification.

As suggested by the reviewer, we compared SMA-miRs’ levels in type II and III patients with 3 SMN2 copies (Figure 3 – figure supplement 3A). The difference was not significant for any miR or the sum. However, the age at sampling was significantly higher in type III (Figure 3 – figure supplement 3B). In our opinion it would be more appropriate to compare age-matched groups to draw any conclusion.

6) There are four patients in the list of type II and III disease with two SMN2 copies. In absence of positive modifier variants (rs121909192 and rs1454173648 also known as NM_017411.3:c.859G>C and NM_017411.3:c.835-44A>G)(both nomenclatures can be included for clarity in the paper) these "outliers" may influence the final score? i.e. PT11 2.70 vs. PT 38 2.12 >6 years (according to data set). Values should be with period not commas, the same applies to the equation.

We have included an additional supplementary table (#2) with individual data for each patient, which has integrated and replaced the previous file “patient_database”. Of the mentioned type II patients, one was twin-sister of a type I male that were previously reported [Pane et al., Neuromuscular Disorders 27 (2017) 890–893]. The 2 other patients were confirmed, both in terms of clinical severity and SMN2 copy number/absence of splicing variants. Regarding the type III patient, due to homonymy, the number of SMN2 copies was erroneously attributed. Indeed, DNA sample of this patient was not available. These data have been amended in figures, table, and equations.

7) What explanation is feasible to explain the correlation differences in > or < 6 years? Can the authors clarify scores >6 years or < 6 years from the same patient?

In supplementary table 2, we have indicated the SMA score obtained with both equations for patients below 6 years. Indeed, the correlation between the scores is very high, since the correlation coefficient is 0.90 and R^2^ is 80.31 (p<10^-5^, n=21, supplementary Figure 5D). These aspects have been pointed in the manuscript.

8) Discussion: neurofilament should be referenced (i.e. Darras BT, et al., Ann Clin Transl Neurol. 2019 Apr 17;6(5):932-944. doi: 10.1002/acn3.779. PMID: 31139691; PMCID: PMC6530526.

See response to item #7 in minor comments.

9) The authors provide additional evidence that muscle is involved in SMA but as they said in Discussion, there is some heterogeneity in the samples. The statement "Taken together, these data suggest that systemic therapeutic approaches increasing SMN levels also in skeletal muscle tissues may provide additional benefits to SMA patients" remains speculative according to what is discussed below this statement about miR-324-5p, miR-451a and miR-181a-5p. I would try to elaborate a statement after the discussion of the role in muscle of the miRNAs detected with a more realistic conclusion.

Thank you very much for the comment. We have rephrased the Discussion accordingly.

10) "More information is (are) available".

Thank you very much for the comment. We have corrected the typo.

11) Please clarify some aspects of the multivariate analysis performed. Selection of variables that are introduced in the multivariate analysis can be done by a full model (the more variables you enter, the better R2 you will have…) or selection of only those variables with significant value in the bivariate linear regression (one by one). The authors state: "Continuous variables were compared by linear correlations. Multivariate analysis was used to correlate severity and molecular parameters (miRNA levels, SMN2 transcripts, SMN2 copy number)". Which variables were significant if compared one by one?

Indeed, even if it has not been reported, in the present study, besides *SMN2*-fl levels, we have also evaluated SMN-del7 and SMN-total transcripts. The inclusion of these variables in the multivariate analyses led to a reduction in both significance and strength of association, since p-value increased and R2 decreased. Thus, we do not agree that by simply adding variables the R2 value increases. Similarly, the single miRNA performed worse than the miR-sum. As reported in the Results, the strongest contribution to the model was not surprisingly provided by SMN2 copy number and SMN2-fl levels.

Besides the contribution of the single variables in the definition of the equation, in our opinion, the correlation of two completely independent variables (decimal SMA classification and SMA-score) is unequivocally strong, up to levels that have never been raised so far in SMA biomarker research. Nonetheless, we fully agree that these data need to be validated in independent cohort, as well as, longitudinal evaluation of SMA-miR levels is required. We remind that we have shown in our previous studies that longitudinal levels of SMN2-fl transcripts are very stable over more than one year (Tiziano et al., 2019). This aspect has been pointed in the Discussion.

12) Please clarify in Table 1 the meaning of P in the column of absolute qPCR.

The p-value has been obtained by comparing miR levels in patients and controls by Mann-Whitney U-test. This aspect has been specified in the table legend and in Methods.

13) The statement "The inclusion of SMA-miRs, age and SMN2-fl levels has almost doubled the accuracy of the severity prediction of SMN2 copy number alone, from about 43% to 82%." should be better discussed. Data of calculations on SMN2 alone (according to the authors apparently is 43%) should be provided because is confusing.

The correlation between SMA Type and SMN2 copy number has been evaluated by a linear regression model (R^2^=52.45%, p<10^-5^, n=41; Figure 5 – figure supplement 1A). This has been specified in the Results and in the statistical analysis section of the Methods. Additionally, we have related SMN2 copies with the decimal SMA classification that has raised the R2 up to 67.00% (Figure 5 – figure supplement 1B). While SMN2 copy number alone provides sufficient predictability in the case of patients with 2 or 4 copies, the relevance in patients with 3 copies is very limited. Indeed, in our cohort, the SMA decimal classification ranges in these patients ranges from 1.9 to 3b. The correlation of the SMA score with the decimal classification in this subgroup of patients (n=21), is significant with a correlation coefficient of 0.55 (p=0.008, R^2^=30.00, Figure 5 – figure supplement 1C).

Looking at the data set Table, in type III there are 13 with 4 copies (54%), 9 with 3 copies (37%) and 2 with 2 copies (8%). In a total of 18 type II patients the score was calculated and 15 (83%) had 3 SMN2 copies as expected in these type of patients. Adding the 4 type I patients the total number of patients with calculated score is 46 (just serum was available in 51 but the score was performed in 46, please mention in the text).

The different biological samples were not available for all patients. As requested, this has been specified in the Methods. Additionally, for each statistical test, the number of samples available has been detailed.

14) I found 7/24 (30%) cases with type III disease with an insufficient SMA-score (less than 2.5 assuming in type II disease according to the score). It is possible to discuss these results?

As reported in supplementary file 2, in the updated version of the equation, the number of type III with low SMA-score is of 1 patient (#26, SMA-score=2.37). This is likely due to the SMN2-fl levels, which appear to be quite low.

15) As the authors mentioned in the last paragraphs, data in newborns (control and SMA) and trajectories with longitudinal data and evolution of the miRNAS should be investigated to ascribe a potential role as a biomarker and to validate the utility of the score.

We agree with the reviewer. Longitudinal data are mandatory for any biomarker study.

16) Even though the variants rs121909192 and rs1454173648 were discarded, a sentence may be included mentioning that investigation of the entire sequence of the SMN2 of patients (perhaps with a reference) may provide additional information about prediction of severity and would be a next step to help improving accuracy of prediction.

As requested, a sentence and a reference have been added.